# Acquisition of auditory discrimination mediated by different processes through two distinct circuits linked to the lateral striatum

**Susumu Setogawa[1,2,3,4,5]\***, **Takashi Okauchi[4]**, **Di Hu[4]**, **Yasuhiro Wada[6]**, **Keigo Hikishima[7]**, **Hirotaka Onoe[8]**, **Kayo Nishizawa[3]**, **Nobuyuki Sakayori[9]**, **Hiroyuki Miyawaki[1,2]**, **Takuma Kitanishi[2]**, **Kenji Mizuseki[1,2]**, **Yilong Cui[4]**, **Kazuto Kobayashi[3]\***

[1]Department of Physiology, Graduate School of Medicine, Osaka Metropolitan University, Osaka, Japan; [2]Department of Physiology, Osaka City University Graduate School of Medicine, Osaka, Japan; [3]Department of Molecular Genetics, Institute of Biomedical Sciences, Fukushima Medical University, Fukushima, Japan; [4]Laboratory for Biofunction Dynamics Imaging, RIKEN Center for Biosystems Dynamics Research, Kobe, Japan; [5]Japan Society for the Promotion of Science, Tokyo, Japan; [6]Laboratory for Pathophysiological and Health Science, RIKEN Center for Biosystems Dynamics Research, Kobe, Japan; [7]Medical Devices Research Group, Health and Medical Research Institute, National Institute of Advanced Industrial Science and Technology, Tsukuba, Japan; [8]Human Brain Research Center, Kyoto University Graduate School of Medicine, Kyoto, Japan; [9]Department of Physiology and Oral Physiology, Graduate School of Biomedical and Health Sciences, Hiroshima University, Hiroshima, Japan

**\*For correspondence:**
setogawa@omu.ac.jp (SS);
kazuto@fmu.ac.jp (KK)

**Competing interest:** The authors declare that no competing interests exist.

## eLife Assessment

This study provides an **important** understanding of the contribution of different striatal subregions, the anterior Dorsal Lateral Striatum (aDLS) and the posterior Ventrolateral Striatum (pVLS), to auditory discrimination learning. The authors have included robust behavior combined with multiple observational and perturbation techniques. The data provided are **convincing** of the relevance of task-related activity in these two subregions during learning.

**Abstract** The striatum, the central hub of cortico-basal ganglia loops, contains functionally heterogeneous subregions distinguished by the topographic patterns of structural connectivity. These subregions mediate various processes of procedural learning. However, it remains unclear when and how striatal subregions engage in the acquisition of sensory stimulus-based decision-making. A neuroimaging of regional brain activity shows that the anterior dorsolateral striatum (aDLS) and posterior ventrolateral striatum (pVLS) in rats are activated in a different temporal pattern during the acquisition phase of auditory discrimination. Chronic and transient pharmacologic manipulations show that the aDLS promotes the behavioral strategy driven by the stimulus-response association while suppressing that by the response-outcome association, and that the pVLS contributes to forming and maintaining the stimulus-response strategy. Electrophysiological recording indicates that subpopulations of aDLS neurons predominantly represent the outcome of specific behaviors at the initial period of discrimination learning, and that pVLS subpopulations encode the beginning and ending of each behavior according to the progress of learning. In addition, other subpopulations of

striatal neurons indicate sustained activation after obtaining reward with distinct patterns reflecting the stimulus-response associations. Our findings demonstrate that aDLS and pVLS neurons integrate the new learning of auditory discrimination in spatiotemporally and functionally different manners.

## Introduction

The neural network connecting the cerebral cortex, basal ganglia, and thalamus forms functionally heterogeneous loops distinguished by the topographic patterns of connectivity (*Alexander and Crutcher, 1990*; *Foster et al., 2021*), and is implicated in various learning processes by the integration of cognitive, sensorimotor, and reward information (*Kim and Hikosaka, 2015*; *Yin and Knowlton, 2006*). The striatum is a key node in the cortico-basal ganglia loops, and its segregated subregions are considered to mediate procedural learning, such as motor skills and instrumental behavior, through parallel processing via different loops (*Kupferschmidt et al., 2017*; *Thorn et al., 2010*; *Yin et al., 2009*). Previous studies in humans and non-human primates have reported that the neural activity within the striatum shifts from the associative subregions (caudate and anterior putamen) to the sensorimotor subregions (posterior putamen) as motor learning progresses (*Doyon et al., 2009*; *Lehéricy et al., 2005*; *Miyachi et al., 2002*; *Miyachi et al., 1997*). In rodents, an instrumental behavior initiates as goal-directed actions and then transits to stimulus-response habits (*Dickinson, 1985*; *O'Hare et al., 2016*), requiring the dorsomedial striatum (DMS; a homologue to the caudate in primates) and the dorsolateral striatum (DLS; a homologue to the putamen in primates), respectively (*Yin et al., 2004*; *Yin et al., 2005*). Based on these results, a model of procedural learning has been proposed, in which the functional dominance changes from the associative to sensorimotor subregions in the striatum during the learning phases (*Redgrave et al., 2010*).

In contrast, deviations from the prior model are evident in other behavioral tasks on decision-making based on external sensory cues. For example, the neural activity in the caudate in humans is associated with learning of a visuomotor association task and the execution of learned behavior (*Seger and Cincotta, 2005*; *Toni and Passingham, 1999*). The tail of the caudate in non-human primates contributes to the automatic, habitual behavior in decision-making tasks using complex visual stimuli (*Kim and Hikosaka, 2013*). In rodents, the DLS is involved in the acquisition and execution of the discrimination task with visual and auditory cues, whereas the DMS mediates the performance of learned behavior (*Featherstone and McDonald, 2004*; *Featherstone and McDonald, 2005*). Moreover, other striatal subregions, such as the posterior striatum and ventrolateral striatum in rodents, are also engaged in auditory discrimination (*Guo et al., 2019*; *Lee et al., 2020*; *Xiong et al., 2015*). Based on these results, a hypothesis has arisen that the acquisition of sensory cue-based decision-making requires wide-range spatiotemporal processing within the striatum. However, the neural mechanism by which the functional circuits linked to striatal subregions change throughout learning processes is not yet fully understood.

To investigate a large-scale striatal reorganization in accordance with the acquisition of external cue-dependent decisions, we conducted a two-alternative auditory discrimination task with rats (*Nishizawa et al., 2012*). Using a small-animal neuroimaging technique, we found that the anterior dorsolateral striatum (aDLS) and posterior ventrolateral striatum (pVLS) are activated to the highest level at the middle and late stages during the acquisition phase of auditory discrimination, respectively. Then, chronic pharmacologic manipulations confirmed that the aDLS and pVLS are necessary for discrimination learning, and the transient manipulations also indicated the function of these subregions in the learning at the corresponding stages. The resultant effects of the manipulations on the behavioral strategy showed that the aDLS promotes the strategy based on the stimulus-response association, while suppressing that based on response-outcome associations. Additionally, the pVLS engages in the formation and maintenance of the stimulus-response association strategy. Electrophysiological data of striatal neurons indicated the presence of aDLS subpopulations, mainly representing the outcome of specific behaviors at the initial period of auditory discrimination and pVLS subpopulations encoding the beginning and ending of behavior in association with the development of the discrimination. There were other subpopulations in the aDLS and pVLS showing long-term increased activity following reward with different patterns of the combination between the stimulus and response. These findings revealed that the two striatal subregions are involved in the acquisition of auditory discrimination through distinct spatiotemporal and functional fashions, challenging

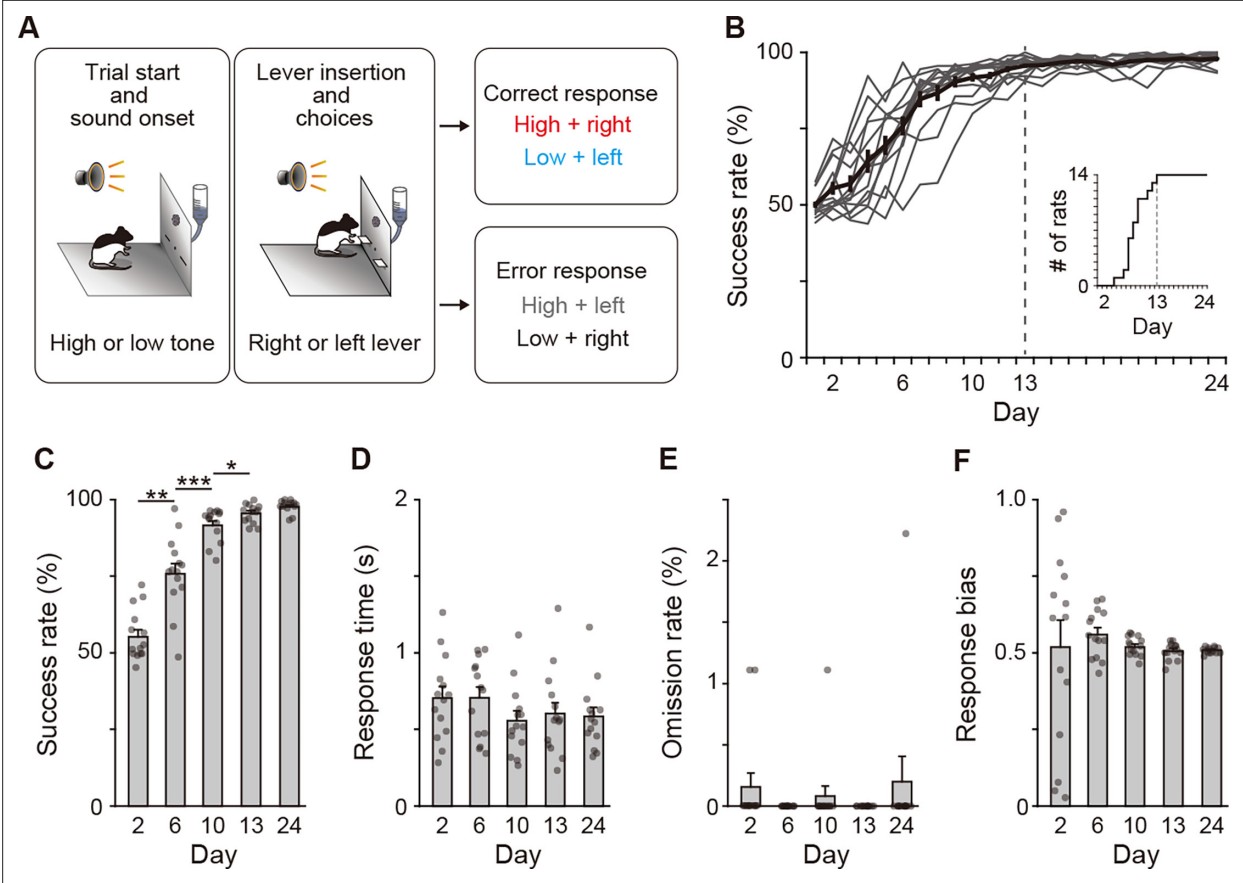

**Figure 1.** Behavioral performance of two-alternative auditory discrimination learning. (**A**) Schematic diagram of the auditory discrimination task. Each trial started with the presentation of a tone instruction cue with the high (10 kHz) or low (2 kHz) frequency. Three seconds later, the room light was illuminated, and two retractable levers were inserted at the same time. The rats were required to press the right and left levers in response to the high and low tones, respectively. (**B**) Learning curve of the auditory discrimination in intact rats (n = 14). Inset indicates the cumulative curve of the number achieving the success rate of more than 90%. (**C**) Success rate at Days 2, 6, 10, 13, and 24 (one-way repeated ANOVA, $F_{[1.897,24.659]}$ = 106.706, p = 1.1 × $10^{-12}$, *post hoc* Bonferroni test, Day 2 vs. Day 6, p = 0.001, Day 6 vs. Day 10, p = 3.6 × $10^{-4}$, Day 10 vs. Day 13, p = 0.040, and Day 13 vs. Day 24, p = 0.080). (**D**) Response time (one-way repeated ANOVA, $F_{[2.203,28.642]}$ = 2.386, p = 0.105). (**E**) Omission rate ($F_{[4,52]}$ = 0.699, p = 0.596). (**F**) Response bias (one-way repeated ANOVA, $F_{[1.129,14.674]}$ = 0.250, p = 0.653). Data are indicated as the mean ± s.e.m., and individual data are overlaid. *p < 0.05, **p < 0.01, and ***p < 0.001.

The online version of this article includes the following source data and figure supplement(s) for figure 1:

**Source data 1.** Behavioral performance of intact rats.

**Figure supplement 1.** Proportions of behavioral strategies during auditory discrimination learning.

**Figure supplement 1—source data 1.** Behavioral strategies of intact rats.

the prior model which proposed a transition from the associative to sensorimotor striatal subregions during learning.

## Results

### Leaning processes of auditory discrimination

We employed a two-alternative auditory discrimination task, in which rats are required to make new associations between a tone instruction cue (high-frequency tone of 10 kHz or low-frequency tone of 2 kHz) and a response (pressing the right or left lever; *Nishizawa et al., 2012*), setting the interval between the stimulus presentation and lever insertion to 3 s (*Figure 1A*). The success rate gradually increased, reaching a plateau around Day 13, which persisted through Day 24 (*Figure 1B*). Based on changes observed in the success rate, we categorized the learning processes into two phases: the

acquisition phase, which lasted until Day 13; and the learned phase, which began after Day 13 and then continued at least until Day 24. In addition, we divided the acquisition phase into three stages depending on the success rate that included the early (<60%), middle (60–80%), and late (80–100%) stages. We chose Days 2, 6, and 10 at the early, middle, and late stages in the acquisition phase, respectively, as well as Days 13 and 24 in the learned phase for statistical comparison of the success rate. There were significant differences between Days 2 and 6, Days 6 and 10, as well as Days 10 and 13, showing no significant difference between Days 13 and 24 (*Figure 1C*). In contrast, the response time and omission rate did not vary among the training days (*Figure 1D and E*), and the variation of response bias decreased along with the progress of auditory discrimination (*Figure 1F*).

## Distinct brain activity patterns in striatal subregions during learning processes

To investigate dynamic changes in regional brain activity in the entire striatum during auditory discrimination in the same animals, we conducted positron emission tomography (PET) for small animals with 2-deoxy-2-[$^{18}$F]fluoro-$_D$-glucose ($^{18}$F-FDG), which measures cerebral glucose metabolism correlated to brain activity (*Cui et al., 2015*; *Kornblum et al., 2000*; *Zimmer et al., 2017*; *Figure 2A*). Before the auditory discrimination task, we performed the single lever press task, in which a lever was pseudo-randomly presented on either the left or right side, and the rats learned to press a lever associated with receiving a reward. This task was used as a control that does not require the decision process based on the auditory stimulus. $^{18}$F-FDG-PET scanning was carried out on Day 4 of the single lever press task, and then during a series of training sessions including the acquisition (Days 2, 6, and 10) and learned (Day 24) phases (*Figure 2B*). The average number of lever presses in the single lever press task exceeded 80 times (81.6 ± 1.3, mean ± s.e.m.) on Day 2 and maintained that level until Day 4 (*Figure 2C*). In the following auditory discrimination task, the success rate gradually increased along with the training days (*Figure 2D*), showing a significant difference among sessions in which we performed the $^{18}$F-FDG-PET scanning (*Figure 2E*). The response time and omission rate were consistent among the days, and the variation of response bias became smaller as the task progressed (*Figure 2E*). These performances were similar to those in the intact rats (compared to *Figure 1C–F*), suggesting that the procedure for $^{18}$F-FDG-PET scans does not affect the acquisition of discrimination.

We investigated the brain regions in which $^{18}$F-FDG uptake significantly increased or decreased during the learning processes, applying a statistical significance threshold (p < 0.001, uncorrected) and an extent threshold, by using a voxel-based statistical parametric analysis (*Cui et al., 2015*). We first analyzed task-related brain activity by comparing $^{18}$F-FDG uptake on either day of the discrimination task with that on Day 4 of the single lever press task (summarized in *Figure 2—source data 2*). In the striatal subregions, there were significant increases in the activity in the unilateral aDLS and the bilateral pVLS on Days 6 and 10, respectively (*Figure 2F and G*). However, no significant change was observed in the DMS during the acquisition phase except for a unilateral decrease on Day 24 (*Figure 2F*). Next, to evaluate the activity related to the progress of learning, $^{18}$F-FDG uptake on Days 6, 10, and 24 was compared with that on Day 2. The learning-dependent brain activity was significantly increased in the bilateral aDLS and the unilateral pVLS on Days 6 and 10, respectively (*Figure 2H*), whereas the activity decreased in the bilateral DMS on Day 24 (*Figure 2H*). This decrease was also detected when $^{18}$F-FDG uptakes were compared between Days 10 and 24 (*Figure 2H*), suggesting the down-regulation of the DMS activity in the learned phase.

To quantitatively validate changes in the brain activity in striatal subregions, we analyzed the amount of $^{18}$F-FDG uptake in the voxel of interest of these subregions (*Figure 2I*). The $^{18}$F-FDG uptake in the bilateral aDLS reached a peak on Day 6, and it decreased during the subsequent days (*Figure 2J*), whereas the uptake in the bilateral pVLS gradually increased along with the progress of learning, showing the maximal value on Day 10 (*Figure 2K*). The uptake in the DMS did not alter during the acquisition phase, and it decreased on Day 24 (*Figure 2L*). These results suggest that the aDLS and pVLS contribute to learning processes in a different temporal pattern during the acquisition phase, whereas the DMS may be engaged to the behavior in the learned phase.

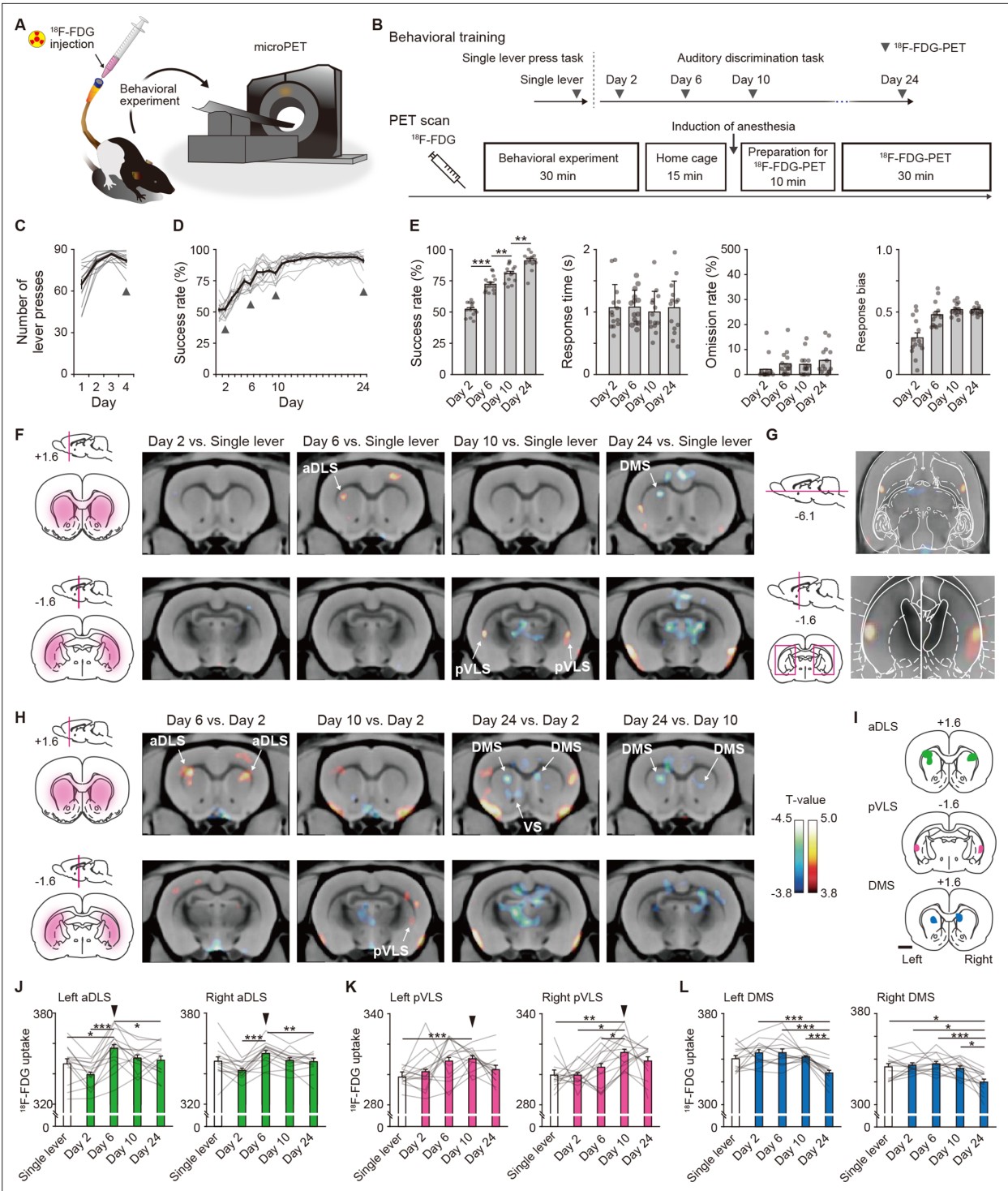

**Figure 2.** Dissociable brain activity patterns during auditory discrimination learning among three striatal subregions. (**A**) Schematic illustration of an awake rat receiving intravenously [18]F-FDG injection through an indwelling catheter attached to the tail. The rat was conducted for the behavioral experiment and then used for microPET imaging. (**B**) Schedule for the behavioral training and [18]F-FDG-PET scan. After [18]F-FDG injection, rats (n = 14 rats) were subjected to the behavioral experiment (30 min) and returned to the home cage (15 min). Ten min before the PET scan started, the rats were anesthetized, and then the scan was started (30 min). (**C, D**) Learning curves of the single lever press task (**C**) and auditory discrimination task (**D**). Arrowheads indicate the PET scan days. (**E**) Behavioral performance on the scan days during the auditory discrimination. Success rate (one-way repeated ANOVA, $F[3,39] = 125.012$, $p = 4.8 \times 10^{-20}$, *post hoc* Bonferroni test; Day 2 vs. Day 6, $p = 5.0 \times 10^{-7}$; Day 6 vs. Day 10, $p = 0.001$; Day 10 vs. Day 24, $p = 0.003$, response time (one-way repeated ANOVA, $F[3,39] = 0.156$, $p = 0.926$), omission rate (one-way repeated ANOVA, $F[3,39] = 1.559$, $p = 0.215$), and response bias (one-way repeated ANOVA, $F[1.779,23.130] = 17.734$, $p = 3.6 \times 10^{-5}$) are shown. (**F**) Representative images of coronal

*Figure 2 continued on next page*

*Figure 2 continued*

sections that compared the brain activity in the single lever press task with the activity on Day 2, 6, 10, or 24 in the discrimination task. Left panels show schematic illustrations of striatal subregions. (**G**) Horizontal (top) and coronal (bottom) images of striatal activation areas shown on Day 10 vs. single lever in (**F**). (**H**) Representative images of coronal section that compared the brain activity on Day 2 with that on Day 6, 10, or 24 in the discrimination task. Color bars indicate the T-values, and a value of 3.8 was used as the threshold corresponding to the uncorrected threshold (p < 0.001). (**I**) Schematic pictures showing voxels of interests for the aDLS, pVLS, and DMS. (**J–L**) Regional $^{18}$F-FDG uptakes in the aDLS (**J**), one-way repeated ANOVA, left aDLS, $F[4,52] = 10.322$, $p = 3.0 \times 10^{-6}$, *post hoc* Bonferroni test; single lever vs. Day 6, p = 0.016; Day 2 vs. Day 6, $p = 6.5 \times 10^{-5}$; Day 2 vs. Day 10, p = 0.019; Day 2 vs. Day 24, p = 0.017; Day 6 vs. Day 24, p = 0.041; right aDLS, $F[4,52] = 7.462$, $p = 7.9 \times 10^{-5}$, *post hoc* Bonferroni test, Day 2 vs. Day 6, $p = 2.9 \times 10^{-4}$; Day 2 vs. Day 10, p = 0.036; Day 6 vs. Day 24, p = 0.005, pVLS (**K**), one-way repeated ANOVA, left pVLS, $F[2.368,30.784] = 4.152$, p = 0.020, *post hoc* Bonferroni test; single lever vs. Day 10, $p = 1.8 \times 10^{-5}$; right pVLS, $F[4,52] = 5.995$, $p = 4.8 \times 10^{-4}$, *post hoc* Bonferroni test, single lever vs. Day 10, p = 0.001; Day 2 vs. Day 10, p = 0.011; Day 6 vs. Day 10, p = 0.032, and DMS (**L**), one-way repeated ANOVA, left DMS, $F[4,52] = 12.836$, $p = 2.4 \times 10^{-7}$, *post hoc* Bonferroni test; Day 2 vs. Day 24, $p = 1.8 \times 10^{-5}$; Day 6 vs. Day 24, $p = 4.8 \times 10^{-4}$; Day 10 vs. Day 24, $p = 5.5 \times 10^{-5}$; right DMS, $F[4,52] = 10.717$, $p=2.0 \times 10^{-6}$, *post hoc* Bonferroni test; single lever vs. Day 24, p = 0.036; Day 2 vs. Day 24, p = 0.011; Day 6 vs. Day 24, $p = 2.6 \times 10^{-4}$; Day 10 vs. Day 24, p = 0.011). Arrowheads indicate the day with the most activations throughout the learning process. Data are indicated as the mean ± s.e.m., and individual data are overlaid. The anteroposterior coordinates from bregma (mm) are shown (**F–I**). Scale bar; 2 mm (**I**). *p < 0.05, **p < 0.01, and ***p < 0.001.

The online version of this article includes the following source data for figure 2:

**Source data 1.** Behavioral performance of rats conducted for microPET imaging and regional brain activity in striatal subregions.

**Source data 2.** Task-related brain activity during the acquisition of auditory discrimination.

## Excitotoxic lesions of the aDLS and pVLS, but not the DMS, disrupt the acquisition of auditory discrimination

To study whether the changed activity of striatal subregions is involved in learning processes, rats received a bilateral injection of solution containing ibotenic acid (IBO; 8 mg/mL, 0.3 µL/site) or phosphate buffered saline (PBS) into the subregions, and we conducted the behavioral study. The range of IBO lesions in the subregions was analyzed by immunostaining with a neuronal marker NeuN after the behavioral tests (*Figure 3A–C*). The number of single lever presses increased similarly along with the training in both the PBS- and IBO-injected groups into the aDLS, pVLS, or DMS along with the training (*Figure 3D–F*). For the aDLS injections, increases in success rate were impaired during the acquisition phase in the IBO group, compared to the PBS group, and the impairments were mainly observed at the middle stage (*Figure 3G*). For the pVLS injection, success rate impairments were observed in the IBO group compared to the PBS group, although the period showing a significant difference was unclear (*Figure 3H*). The response time in the aDLS-lesioned group did not alter during the discrimination training, whereas that in the pVLS-lesioned group was continuously prolonged (*Figure 3—figure supplement 1A, B*), and the omission rate and response bias were similar between the lesioned and the corresponding control groups (*Figure 3—figure supplement 1D, G* for aDLS lesion, and *Figure 3—figure supplement 1E, H* for pVLS lesion). In contrast, for the DMS injection, the success rate in the discrimination task was comparable between the PBS and IBO groups (*Figure 3I*). The DMS lesion lengthened the response time through the learning (*Figure 3—figure supplement 1C*), whereas it did not alter the omission rate and response bias (*Figure 3—figure supplement 1F, I*). These results indicate that the aDLS and pVLS are necessary for the acquisition of auditory discrimination, and that the aDLS appears to be mainly involved in the learning process at the middle stage. The results also reveal that the DMS contributes to the execution, but not learning, of discriminative behavior.

Previous studies have reported that the dorsal striatum is involved in both behavioral strategies based on the stimulus-response association and the response-outcome association and during instrumental learning (*Dickinson, 1985*; *O'Hare et al., 2016*). To assess the impact of striatal lesions on the behavioral strategies, we analyzed the proportions of responses attributed to two strategies in all responses of each session. One is the 'win-shift-win' strategy, which is considered to reflect the behavioral strategy based on the stimulus-response association (*Bergstrom et al., 2018*; *Packard et al., 1989*). In this strategy, after a correct response in the previous trial, the rats press the opposite lever in the current trial in response to a shift of the instruction cue, resulting in the correct response. Another strategy is the 'lose-shift-lose' strategy, which is considered to appear as a consequence of the behavioral strategy based on the response-outcome association (*Bergstrom et al., 2018*). In this strategy, after an error response in the previous trial, the rats press the opposite lever in the current trial, despite a shift of the instruction cue, leading to another error response. In the control group

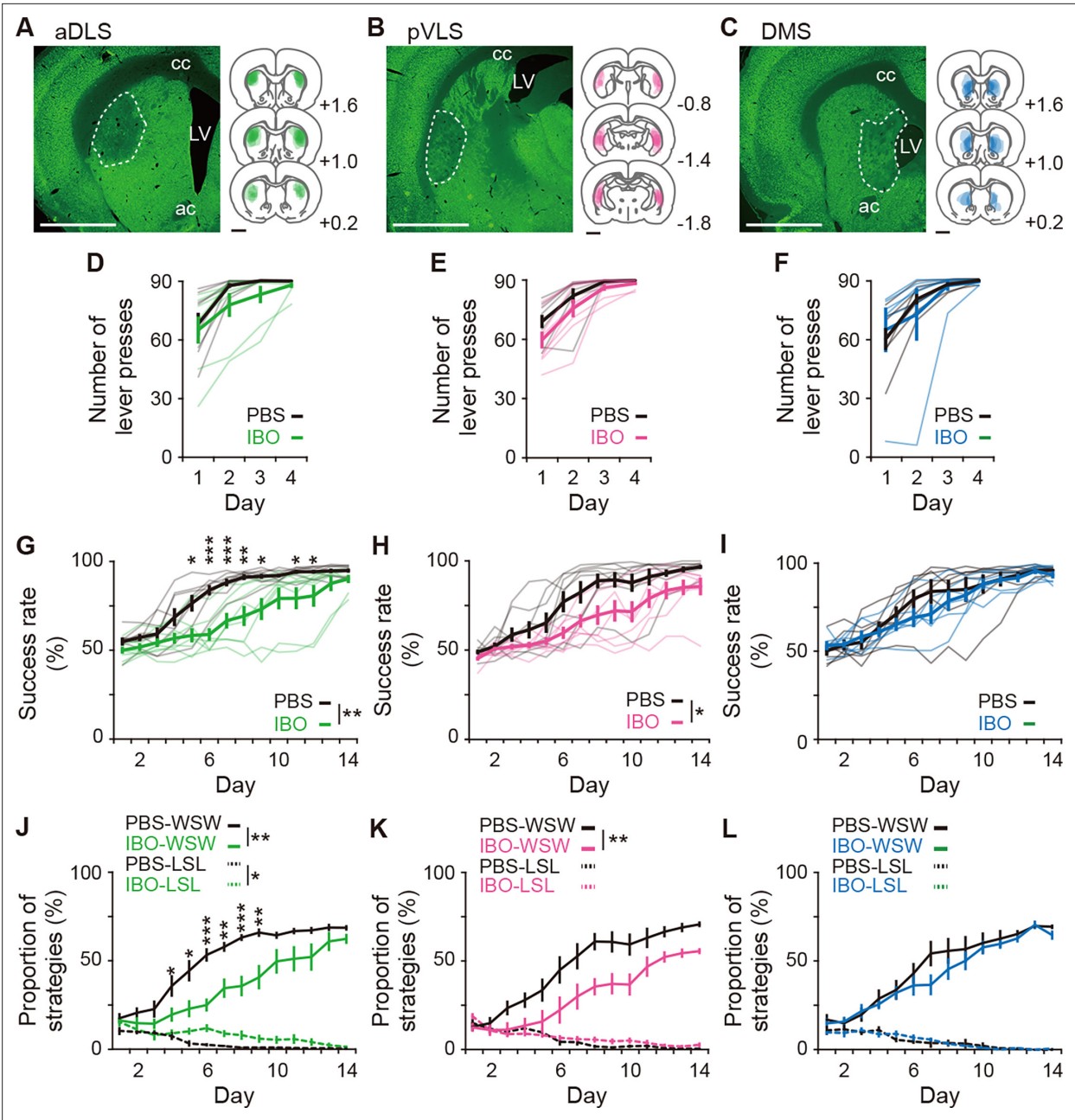

**Figure 3.** Impacts of excitotoxic lesion of striatal subregions on the acquisition of auditory discrimination. Rats were given intracranial injection of PBS or IBO solution into the aDLS (n=8 for each injection), pVLS (n = 8 for each injection), or DMS (n = 7 for PBS injection, and n = 6 for IBO injection) before the start of the single lever press task. (**A–C**) Representative images of NeuN immunostaining and individual schematic illustrations showing lesioned area in the aDLS (**A**), pVLS (**B**), or DMS (**C**). Dotted lines in the images indicate the range of the lesioned area. ac, anterior commissure; cc, corpus callosum; and LV, lateral ventricle. (**D–F**) Learning curves during the single lever press task in the groups injected into the aDLS (**D**), two-way repeated ANOVA, group, $F[1,14] = 1.275$, p = 0.278, day, $F[1.512,21.171] = 26.281$, p = $7.0 \times 10^{-6}$, group × day, $F[1.512,21.171] = 0.834$, p = 0.418, pVLS (**E**), two-way repeated ANOVA, group, $F[1,14] = 2.542$, p = 0.133, day, $F[1.547,21.658] = 40.183$, p = $2.2 \times 10^{-7}$, group × day, $F[1.547,21.658] = 0.953$, p = 0.380, or DMS (**F**), two-way repeated ANOVA, group, $F[1,11] = 0.025$, p = 0.876, day, $F[1.205,13.252] = 14.818$, p = 0.001, group ×day, $F[1.205,13.252] = 0.610$, p = 0.478. (**G–I**) Learning curves during the auditory discrimination task in the groups injected into the aDLS (**G**), two-way repeated ANOVA, group, $F[1,14] = 11.578$, p = 0.004, day, $F[2.805,39.274] = 60.733$, p = $1.8 \times 10^{-14}$, group × day, $F[2.805,39.274] = 3.863$, p = 0.018, simple main effect, PBS vs. IBO, Day 1, p = 0.140, Day 2, p = 0.085, Day 3, p = 0.284, Day 4, p = 0.068, Day 5, p = 0.011, Day 6, p = $9.5 \times 10^{-5}$, Day 7, p = 0.001, Day 8, p = 0.002, Day 9, p = 0.012, Day 10, p = 0.059, Day 11, p = 0.034, Day 12, p = 0.046, Day 13, p = 0.136, Day 14, p = 0.096, pVLS (**H**), two-way repeated ANOVA, group, $F[1,14] = 7.221$, p = 0.018, day, $F[3.646,51.041] = 51.794$, p = $9.2 \times 10^{-17}$, group × day, $F[3.646,51.041] = 1.612$, p = 0.190, or DMS (**I**), two-way repeated ANOVA, group, $F[1,14] = 0.469$, p = 0.507, day, $F[2.677,29.452] = 48.860$, p = $3.9 \times 10^{-11}$, group × day, $F[2.677,29.452] = 1.094$, p = 0.362. (**J–L**) Changes in the proportions of the win-shift-win or lose-shift-lose strategies through the discrimination learning in the groups treated into the aDLS (**J**), two-way

*Figure 3 continued on next page*

*Figure 3 continued*

repeated ANOVA, group, $F[1,14] = 13.451$, p = 0.003, day, $F[3.048,42.673] = 54.777$, p = $9.1 \times 10^{-15}$, group × day, $F[3.048,42.673] = 3.288$, p = 0.029, simple main effect, PBS vs. IBO, Day 1, p = 0.728, Day 2, p = 0.225, Day 3, p = 0.165, Day 4, p = 0.044, Day 5, p = 0.015, Day 6, p = $1.1 \times 10^{-4}$, Day 7, p = 0.002, Day 8, p = $5.1 \times 10^{-4}$, Day 9, p = 0.004, Day 10, p = 0.060, Day 11, p = 0.053, Day 12, p = 0.081, Day 13, p = 0.198, Day 14, p = 0.075 for win-shift-win; group, $F[1,14] = 5.039$, p = 0.041, day, $F[3.866,54.119] = 14.578$, p = $5.2 \times 10^{-8}$, group × day, $F[3.866,54.119] = 2.471$, p = 0.057 for lose-shift-lose, pVLS (**K**), two-way repeated ANOVA, group, $F[1,14] = 9.251$, p = 0.009, day, $F[3.206,44.881] = 47.168$, p = $2.7 \times 10^{-14}$, group × day, $F[3.206,44.881] = 1.862$, p = 0.146 for win-shift-win; group, $F[1,14] = 0.762$, p = 0.397, day, $F[4.773,66.815] = 24.018$, p = $1.9 \times 10^{-13}$, group × day, $F[4.773,66.815] = 1.338$, p = 0.260 for lose-shift-lose, or DMS (**L**), two-way repeated ANOVA, group, $F[1,14] = 0.703$, p = 0.420, day, $F[3.052,33.568] = 48.424$, p = $2.3 \times 10^{-12}$, group × day, $F[3.052,33.568] = 0.846$, p = 0.480 for win-shift-win; group, $F[1,14] = 0.002$, p = 0.965, day, $F[3.442,37.866] = 13.787$, p = $1.0 \times 10^{-6}$, group × day, $F[3.442, 37.866] = 0.581$, p = 0.654 for lose-shift-lose. Data are indicated as the mean ± s.e.m., and individual data are overlaid (except for panels **J–L**). The anteroposterior coordinates from bregma (mm) are shown (**A–C**). Scale bars; 2 mm (**A–C**). *p < 0.05, **p < 0.01 and ***p < 0.001. Abbreviations: WSW, win-shift-win; and LSL, lose-shift-lose.

The online version of this article includes the following source data and figure supplement(s) for figure 3:

**Source data 1.** Behavioral performance and strategies of IBO-lesioned rats.

**Figure supplement 1.** Effects of excitotoxic lesion on the reaction time, omission ratio, and response bias.

**Figure supplement 1—source data 1.** Reaction time, omission ratio, and response bias of IBO-lesioned rats.

**Figure supplement 2.** Effects of excitotoxic lesion on feeding behavior and locomotion.

**Figure supplement 2—source data 1.** Feeding behavior and locomotion of IBO-lesioned rats.

into either of three striatal subregions, the proportions of win-shift-win and lose-shift-lose strategies increased and decreased, respectively, during the acquisition phase (*Figure 3J–L*). The changes in the proportion of the two strategies were similar to those in the intact rats (*Figure 1—figure supplement 1*). For the aDLS injection, the increases in the win-shift-win proportion were markedly impaired in the IBO group compared to the PBS group, with significant differences mainly at the middle stage, and the decrease in the lose-shift-lose proportion was delayed compared to the same group (*Figure 3J*). For the pVLS injection, the increasing win-shift-win proportion was also impaired in the IBO group compared to the PBS group, although the change in the lose-shift-lose proportion was comparable between the two groups (*Figure 3K*). In contrast, for the DMS lesion, both win-shift-win and lose-shift-lose proportions in the IBO group did not show significant differences from the corresponding values in their control group (*Figure 3L*). These results indicate that the aDLS promotes the win-shift-win strategy through stimulus-response association and suppresses the lose-shift-lose strategy rooted in response-outcome association, and that the pVLS selectively facilitates the win-shift-win strategy acquisition. Thus, these two subregions not only play roles in the acquisition of auditory discrimination but also control distinct behavioral strategies during the learning processes.

After the completion of discrimination training, we tested feeding behavior and locomotion of rats with aDLS and pVLS lesions. There was no difference in the amount of food intake and locomotor activity in the open field in both lesioned groups (*Figure 3—figure supplement 2A–D*), suggesting that learning deficits following excitotoxic lesions of the striatal subregions cannot be attributable to abnormalities in these general motor behaviors.

## Inhibiting aDLS and pVLS activity impairs discriminative behavior at different stages of the acquisition phase

Our [18]F-FDG-PET imaging study indicated that the brain activity in the aDLS and pVLS was increased in a different temporal pattern during the acquisition phase of auditory discrimination. To examine whether neuronal activity in striatal subregions is linked with the processes at different stages, we temporarily inhibited the activity in each subregion by the treatment with the GABA$_A$ receptor agonist muscimol (MUS). In a pilot study, we tested the effect of the bilateral injection of several doses of MUS (0.1, 0.2, and 0.5 μg/μL) into the aDLS or pVLS on the single lever press behavior and confirmed that the administration of the lowest dose of MUS did not change the frequency of single lever presses as compared to the saline (SAL) injection (*Figure 4—figure supplement 1A, B*). The lowest dose of MUS into the two striatal subregions did not affect the amount of food intake (*Figure 4—figure supplement 1C, D*). In addition, to achieve the stage-specific inactivation of the aDLS or pVLS neurons, rats were given bilateral injections of MUS (0.1 μg/μL, 0.25 μL per site) or SAL at three different timings during a series of the training. The first injection was conducted on Day 2 to target the early stage. The

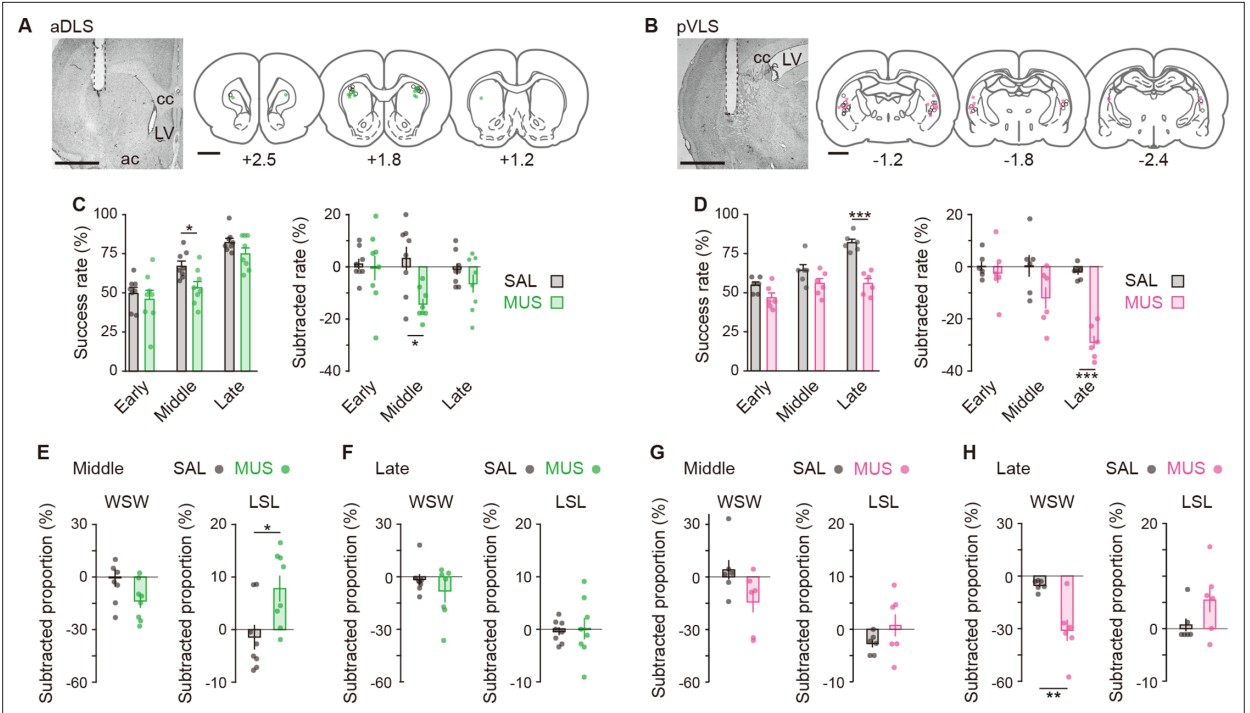

**Figure 4.** Influences of transient inhibition of striatal subregions at different timings on the performance of auditory discrimination. Rats received intracranial injection of SAL or MUS solution into the aDLS (n = 8 for each injection) or pVLS (n = 6 for each injection). (**A, B**) Representative images of cresyl violet staining and schematic illustrations showing placement sites of the tip of guide cannula in the aDLS (**A**) or pVLS (**B**). Dotted lines in the images indicate the position of the cannula placement. ac, anterior commissure; cc, corpus callosum; and LV, lateral ventricle. (**C, D**) Effects of transient striatal inhibition on the performance. For aDLS inhibition (**C**), success rate (left, two-way repeated ANOVA, stage, $F[2,28] = 28.708$, p = $1.7 \times 10^{-7}$, group, $F[1,14] = 8.840$, p = 0.010, stage × group, $F[1,14] = 0.757$, p = 0.478; unpaired Student's t-test, early, $t[14] = 0.558$, p = 0.586; middle, $t[14] = 2.764$, p = 0.015; Welch's t-test, late, $t[12.010] = 1.617$, p = 0.132, p = 0.016 after a Bonferroni correction method) and subtracted success rate on day N-1 from that on day N (right, two-way repeated ANOVA, stage, $F[2,28] = 1.231$, p = 0.307, group, $F[1,14] = 10.797$, p = 0.005, stage × group, $F[2,28] = 2.360$, p = 0.113; unpaired Student's t-test, early, $t[14] = 0.285$, p = 0.780, late, $t[14] = 1.271$, p = 0.225, Welch's t-test, middle, $t[9.603] = 3.331$, p = 0.008, p = 0.016 after a Bonferroni correction method). For pVLS inhibition (**D**), success rate (left, two-way repeated ANOVA, stage, $F[2,28] = 17.642$, p = $3.8 \times 10^{-5}$, group, $F[1,10] = 43.942$, p = $5.9 \times 10^{-5}$, stage × group, $F[2,20] = 4.729$, p = 0.012; simple main effect, early, p = 0.055, middle, p = 0.109, late, p = $4.2 \times 10^{-5}$) and subtracted success rate on day N-1 from that on day N (right, two-way repeated ANOVA, stage, $F[2,20] = 12.105$, p = $3.6 \times 10^{-4}$, group, $F[1,10] = 16.310$, p = 0.002, stage × group, $F[2,20] = 8.500$, p = 0.002; simple main effect, early, p = 0.595, middle, p = 0.085, late, p = $3.0 \times 10^{-6}$). (**E, F**) Subtracted proportion of the win-shift-win or lose-shift-lose strategies in the groups injected into the aDLS at the middle stage (**E**), unpaired Student's t-test, $t[14] = 2.038$, p = 0.061 for win-shift-win, and $t[14] = 2.714$, p = 0.017 for lose-shift-lose and the late stage (**F**), unpaired Student's t-test, $t[14] = 0.898$, p = 0.384 for win-shift-win, and $t[14] = 0.226$, p = 0.824 for lose-shift-lose. (**G, H**) Subtracted proportion of the win-shift-win or lose-shift-lose strategies in the groups injected into the pVLS at the middle stage (**G**), unpaired Student's t-test, $t[10] = 1.924$, p = 0.083 for win-shift-win, and Welch's t-test, $t[6.095] = 1.364$, p = 0.221 for lose-shift-lose and the late stage (**H**), unpaired Student's t-test, $t[10] = 3.629$, p = 0.005 for win-shift-win, and $t[10] = 1.577$, p = 0.146 for lose-shift-lose. Data are indicated as the mean ± s.e.m., and individual data are overlaid. The anteroposterior coordinates from bregma (mm) are shown (**A, B**). Scale bars; 2 mm (**A, B**). *p < 0.05, **p < 0.01, and ***p < 0.001. Abbreviations: WSW, win-shift-win; and LSL, lose-shift-lose.

The online version of this article includes the following source data and figure supplement(s) for figure 4:

**Source data 1.** Behavioral performance and strategies of MUS-injected rats.

**Figure supplement 1.** Effects of bilateral MUS injections into the aDLS and pVLS on the single lever press task and feeding behavior.

**Figure supplement 1—source data 1.** Single lever press task and feeding behavior of MUS-injected rats.

**Figure supplement 2.** Performance of the auditory discrimination task on Day N-1 of rats used for the injection into striatal subregions.

**Figure supplement 2—source data 1.** Behavioral performance on Day N-1 of MUS-injected rats.

**Figure supplement 3.** Proportion of the behavioral strategies on Day N-1 of rats used for the striatal injection.

**Figure supplement 3—source data 1.** Behavioral strategies on Day N-1 of MUS-injected rats.

second and third injections were conducted on the day after the success rate had reached 60% and 80% for the first time through the training, respectively, to target the middle or late stage. After the injections, the rats were used for the behavioral test. Placement locations of the injection cannula into the striatal subregions were assessed by cresyl violet staining after the behavioral tests (*Figure 4A and B*). The success rates on the day before the injection (Day N-1) at the early, middle, and late stages were similar between the two groups that received the bilateral injections of MUS or SAL (*Figure 4— figure supplement 2A, B*). Rats were then administered injections into the aDLS on the next day (Day N), and the success rate in the MUS-injected groups significantly decreased at the middle stage compared to the corresponding SAL-injected groups, but not at the early and late stages (*Figure 4C*, left). In contrast, the success rate in the MUS groups into the pVLS significantly decreased at the late stage, but not at the early and middle stages, compared with the respective SAL groups (*Figure 4D*, left). To normalize the variation of learning speed in the individual animals, we calculated the value by subtracting the success rate on Day N-1 from that on Day N in the individual animals. There were significant reductions in the subtracted rate at the middle stage in the MUS-injected group into the aDLS (*Figure 4C*, right) and at the late stage in the MUS-injected group into the pVLS (*Figure 4D*, right). These results indicate that the aDLS and pVLS mainly function at the middle and late stages, respectively, in the acquisition phase of auditory discrimination, supporting the results obtained from our imaging study.

Next, we analyzed the influence of temporal striatal inactivation on the behavioral strategy at the middle and late stages of the acquisition phase. We calculated the proportions of the win-shift-win and lose-shift-lose strategies and subtracted the values of these proportions on Day N-1 from those on Day N. The proportion of both strategies on Day N-1 was comparable between the MUS- and SAL-injected groups (*Figure 4—figure supplement 3A, B*). In the aDLS, the subtracted proportion of the lose-shift-lose strategy at the middle stage significantly increased in the MUS group as compared to the SAL group, although the subtracted proportion of the win-shift-win strategy tended to decrease in the MUS group as compared to the control group (*Figure 4E*), and the subtracted proportions of the win-shift-win or lose-shift-lose strategy at the late stage were similar between both groups (*Figure 4F*). In the pVLS, the subtracted proportions of the win-shift-win and lose-shift-lose strategies at the middle stage were similar between the MUS and SAL groups (*Figure 4G*), and the subtracted win-shift-win proportion at the late stage significantly decreased in the MUS group as compared to the SAL group, although the subtracted lose-shift-lose proportion was comparable between the two groups (*Figure 4H*). Our results show that transient inhibition of aDLS activity at the middle stage affects the lose-shift-lose strategy and has a subtle effect on the win-shift-win strategy, supporting that the aDLS mediates the acquisition process of auditory discrimination at the middle stage at least partly through the suppression of the behavior based on the response-outcome association. Although the impact of inhibition on the win-shift-win strategy was subtle, excitotoxic lesion of the aDLS influenced the win-shift-win strategy in addition to the lose-shift-lose strategy (see *Figure 3J*), suggesting that the aDLS acts to enhance the auditory discrimination by promoting the stimulus-response association, together with the suppression of the response-outcome association. The discrepancy in the extent of changes in the win-shift-win strategy between the two treatments may be generated from the difference in the timing or sustainability of drug inhibition of the striatal subregion. In addition, our results show that pVLS inactivation at the late stage selectively impairs the win-shift-win strategy, supporting the contribution of the pVLS to the process via progression of the stimulus-response association.

## Neuronal activity in the aDLS and pVLS during the acquisition phase

To examine the firing activity in striatal subregions during the acquisition phase of auditory discrimination, we simultaneously recorded neuronal activity in the aDLS and pVLS at the unilateral (left) side by using a multi-unit recording procedure (*Figure 5A*). Using immunostaining for tyrosine hydroxylase (TH) after the recordings, we verified the location of electrodes within the striatal subregions (*Figure 5B*). After the single lever press task, we conducted the auditory discrimination task for the recordings. In correct trials, the delay period (0.5 ± 0.2 s) was inserted between a lever press and a sound indicating the reward to distinguish neuronal activity related to the two events (*Figure 5C*). The behavioral data in individual rats were divided into three grouped sessions corresponding to the early, middle, and late stages, based on a sigmoid function analysis (*Bathellier et al., 2013*) of the success rate in the task. The success rates in the grouped sessions were 52.7 ± 0.8%, 61.6 ± 0.7%, and 85.3 ±

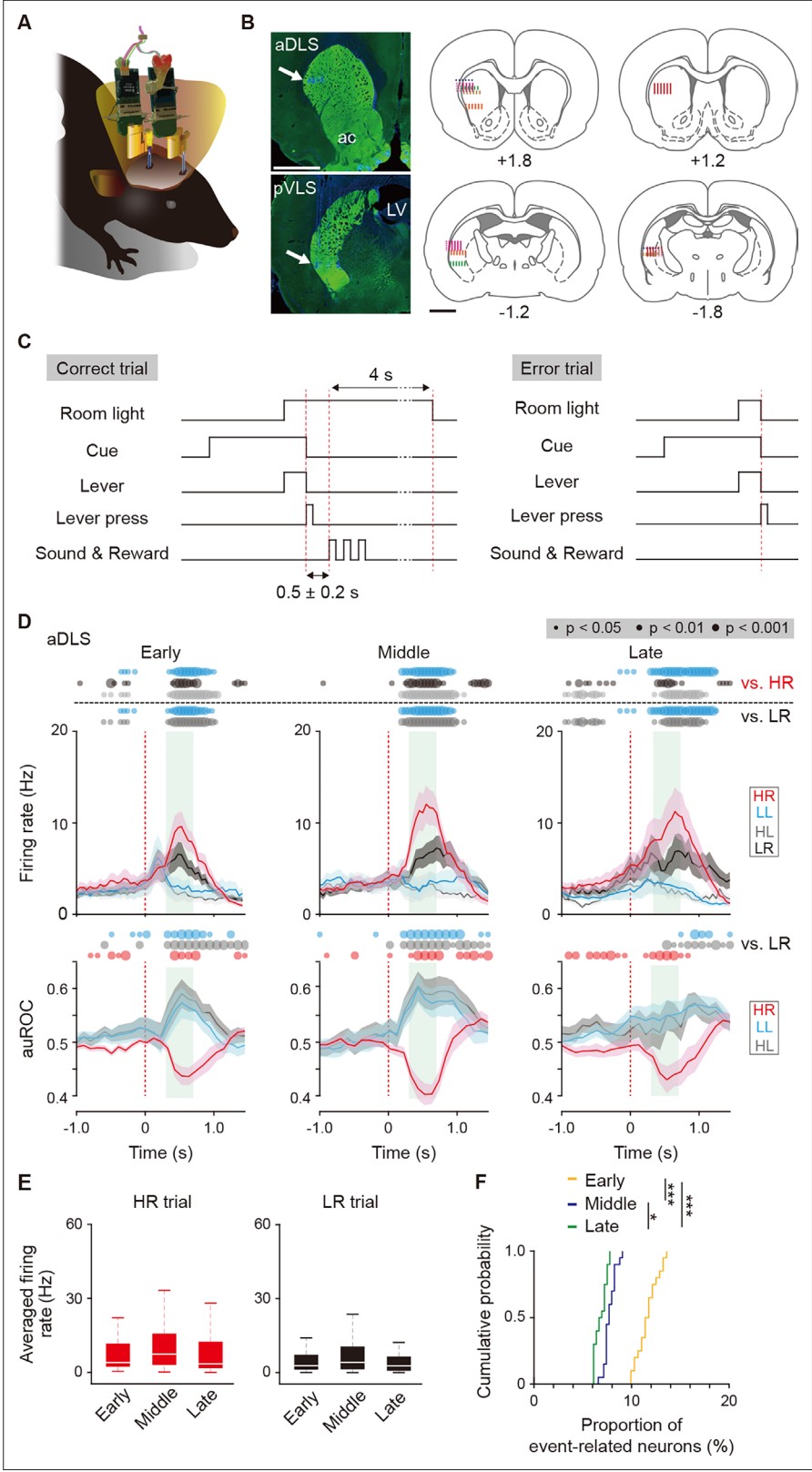

**Figure 5.** Multi-unit recording of neurons in striatal subregions and firing activity related to the behavioral outcome of reward sound-related HR type neurons in the aDLS. (**A**) Schematic illustration of a freely moving rat used for simultaneous multi-unit recordings in the aDLS and pVLS. Six rats were used for the following analysis. (**B**) Representative images of TH immunostaining, showing positions of electrode tips (arrows) in the aDLS and

*Figure 5 continued*

pVLS (left rows). The recording sites estimated by electrode tracks and electrical marks in individual rats are shown (center and right rows). ac, anterior commissure; LV, lateral ventricle. (**C**) Sequence of some events in correct and error trials. The delay periods were pseudorandomly added between a correct lever press and the reward sound (0.5 ± 0.2 s). The room light turned off the extended time (4 s) after the correct response or immediately after the error response. (**D**) Mean firing rate (top rows) and auROC values (bottom rows) of reward sound-related HR type neurons in the aDLS at the early (n = 43 neurons), middle (n = 37 neurons), and late (n = 44 neurons) stages. These data are aligned to the choice response. Green shadows show the period when the reward sound is presented (500 ± 200 ms after the lever press). Time bins with significant differences between the mean firing rate in the HR or LR trial and any of the rates in the other three trials (Wilcoxon signed rank test) or the distribution of auROC values and 0.5 (Wilcoxon rank-sum test) are represented by the circles at the top. (**E**) Averaged firing rate during the reward sound period in the HR and LR trials at the early, middle, and late stages (Kruskal-Wallis test: HR trial, $\chi^2$ = 3.028, p = 0.220; and LR trial, $\chi^2$ = 1.249, p = 0.536). (**F**) Cumulative probability of the proportion of reward sound-related HR type neuron number at the three stages (Kruskal-Wallis test, $\chi^2$ = 45.378, p = 1.4 × 10$^{-10}$; *post hoc* Tukey-Kramer test, p = 7.1 × 10$^{-5}$ for early vs. middle, p = 1.0 × 10$^{-9}$ for early vs. late, and p = 0.040 for middle vs. late). Data are indicated as the mean ± s.e.m. (**D**) or the median and quartiles with the maximal and minimal values (**E**). The anteroposterior coordinates from bregma (mm) are shown (**B**). Scale bars; 2 mm (**B**). *p < 0.05 and ***p < 0.001.

The online version of this article includes the following source data and figure supplement(s) for figure 5:

**Source data 1.** Firing activity of reward sound-related HR type neurons in the aDLS.

**Figure supplement 1.** Behavioral performance of rats used for the multi-unit recording experiment.

**Figure supplement 1—source data 1.** Behavioral performance of rats used for the multi-unit recording.

**Figure supplement 2.** Properties of identified neurons in the multi-unit recording experiment.

**Figure supplement 2—source data 1.** Numbers of identified neurons in the multi-unit recording.

**Figure supplement 3.** Changes in event-related firing activity of HR and LL type neurons during the acquisition phase of the auditory discrimination task.

**Figure supplement 3—source data 1.** Proportions of event-related neurons during the auditory discrimination task.

**Figure supplement 4.** Firing activity of reward sound-related LL type neurons in the aDLS at the three learning stages.

**Figure supplement 4—source data 1.** Firing activity of reward sound-related LL type neurons in the aDLS.

**Figure supplement 5.** Firing activity of reward sound-related HR and LL type neurons in the pVLS at the three learning stages.

**Figure supplement 5—source data 1.** Firing activity of reward sound-related neurons in the pVLS.

1.7% (mean ± s.e.m.) for the early, middle, and late stages, respectively, and the rate was significantly increased throughout the three stages (***Figure 5—figure supplement 1A***). However, the response time, omission rate, and response bias were similar among the stages (***Figure 5—figure supplement 1B–D***). Thus, these stages corresponded to the timing window defined in the acquisition phase of the transient MUS inhibition experiment of neuronal activity.

We identified 1062 and 423 well-isolated neurons from the aDLS and pVLS, respectively, across the three stages (***Figure 5—figure supplement 2A***) and focused on the major population of striatal neurons, putative medium spiny neurons (MSNs; ***Figure 5—figure supplement 2B***). The numbers of MSNs were 96% and 91% of the total numbers of identified neurons in the aDLS and pVLS, respectively (***Figure 5—figure supplement 2C***). The neuron numbers at the early, middle, and late stages were 295, 364, and 360 neurons in the aDLS, and 87, 110, and 190 neurons in the pVLS, respectively. Based on the combinatorial pattern of the tone instruction cue and lever press in our discrimination task, we categorized the electrophysiological data into four trial types: (1) high-frequency tone/right lever press (HR) and (2) low-frequency tone/left lever press (LL) as correct responses; and (3) high-frequency tone/left lever press (HL) and (4) low-frequency tone/right lever press (LR) as error responses (see ***Figure 1A***). We identified HR or LL type neurons showing significant changes in firing rate related to the cue onset, choice response, reward sound, or first licking as compared to the baseline firing rate in each of two trials (Mann-Whitney *U* test, p<0.01). These neurons were further divided into two groups based on increased or decreased activity (***Figure 5—figure supplement 3A and B***). In the following

analyses, we focused on the HR and LL type neurons with increased event-related activity to explore the firing patterns of neurons underlying behavioral processes of auditory discrimination.

## Subpopulations of aDLS neurons, but not of pVLS neurons, show firing activity related to the behavioral outcome

Previous studies have reported that neurons in the dorsal striatum show the outcome-related activity in the instrumental conditioning tasks (*Guo et al., 2019*; *Nonomura et al., 2018*). We thus investigated whether aDLS and pVLS neurons exhibit a response related to the behavioral outcome during the acquisition of auditory discrimination. In the aDLS, the firing activity of reward sound-related HR type neurons showed a higher extent in the increase during a period when the reward sound is presented in the correct responses, in the HR trial as compared with the LL, HL, or LR trial throughout the three stages (*Figure 5D*, top rows). The rate of these neurons also moderately increased during the same period in the LR trial compared to the LL or HL trial across the three stages, although the extent of the increase was lower than that in the HR trial (*Figure 5D*, top rows). The area under the receiver operating characteristic (auROC) values revealed a significant difference of the HR trial from the LR trial during the reward sound period, whereas the values of LL and HL trials indicated a significant difference from the LR trial in the opposite direction during the same period (*Figure 5D*, bottom rows). The averaged firing rate during the reward sound period in the HR or LR trial appeared to modestly increase at the middle stage, but no significant difference existed among the stages (*Figure 5E*). To directly compare the number of neurons showing event-related activity among different stages, we calculated the number of reward sound-HR type neurons by matching the trial number at each stage (*Figure 5F*). The proportion decreased progressively from the early to middle stages and to the late stage. In addition, the firing rate of reward sound-related LL type neurons exhibited a higher increase in the LL trial and a modest increase in the HL trial, while the averaged firing rate during the reward sound period was similar among the three stages (*Figure 5—figure supplement 4A, B*). Furthermore, the proportion of reward sound-LL neurons reduced progressively through the stages (*Figure 5—figure supplement 4C*). These results indicate that the aDLS contained neurons related to the outcome of choice, showing a greater increase in the firing activity for the correct responses (HR and LL trials) than the error responses (HL and LR trials), and that the proportion of these subpopulations was gradually reduced through the progress of learning. In contrast, during the reward sound period at all stages, the firing rate and auROC values of reward sound-HR and reward sound-LL type neurons in the pVLS revealed no significant differences across the four kinds of trial (*Figure 5—figure supplement 5A, B*), indicating that the firing activity of pVLS neurons did not appear to be associated with the behavioral outcome.

## Subpopulations of aDLS and pVLS neurons show sustained firing activity after the reward in different manners

The dorsal striatum is known to demonstrate a response of neuronal activity after the positive outcome in the instrumental behaviors (*Smith and Graybiel, 2016*; *Williams and Eskandar, 2006*). We thus examined whether aDLS and pVLS neurons show the response after obtaining the reward during the acquisition phase of auditory discrimination. In the aDLS, the firing rate of the first licking-related HR type neurons indicated a sustained elevation after the first licking in the HR trial as compared with any of the other three trials across the early, middle, and late stages (*Figure 6A*, top rows). The auROC values revealed a significant difference of the LL, HL, or LR trial from the HR trial across the three stages (*Figure 6A*, middle rows). In addition, we calculated the averaged firing rate and increased firing period, defined as the total number of time bins above 3 s.d. of the baseline firing rate for 5 s after the first licking in the HR trial. The averaged firing rate was comparable between the early and middle stages, but the rate was significantly lower at the late stage compared to either of the other two stages (*Figure 6B*). The increased firing period was also comparable between the early and middle stages, showing a significant reduction at the late stage (*Figure 6C*). The proportions of first licking-HR type neurons were similar between the early and late stages, although they showed a slight decrease at the middle stage (*Figure 6D*). There were only a few neurons with significant correlations between the firing rate and licking performance (*Figure 6E*). In addition, the firing rate of the first licking-related LL type neurons exhibited the largest increase in the LL trial at the three stages, showing a tendency to reduce the average firing rate and increased firing period after the first

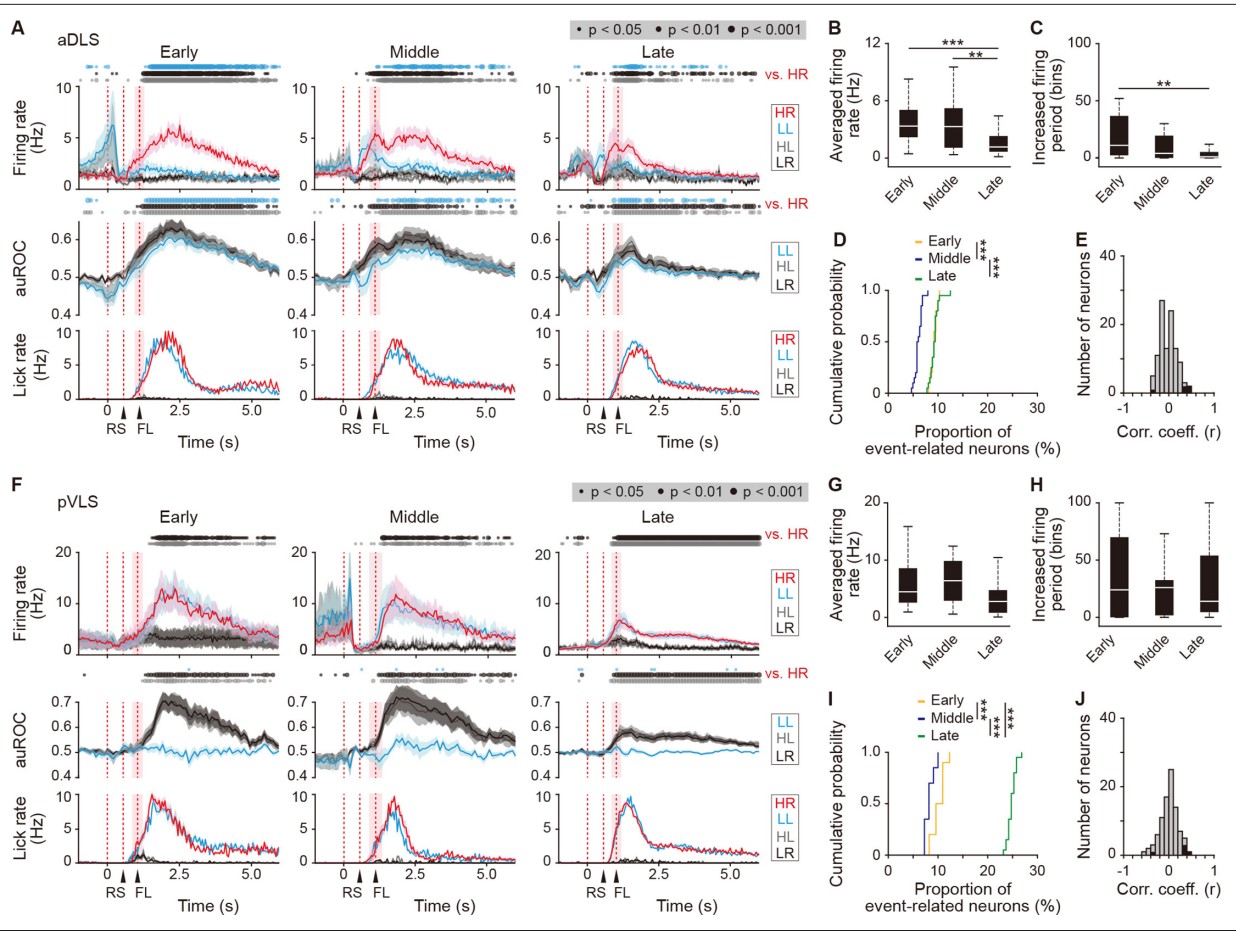

**Figure 6.** Sustained activity after the reward of first licking-related HR type neurons in the aDLS and pVLS. (**A**) Firing rate (top rows), auROC (middle rows), and licking rate (bottom rows) after the reward of first licking-related HR type neurons in the aDLS at the early (n = 27 neurons), middle (n = 31 neurons), and late (n = 49 neurons) stages. These data are aligned to the choice response. Timings of the reward sound and first licking are shown. Time bins with significant differences between the mean firing rate in the HR trial and any of the rates in the other three trials (Wilcoxon signed rank test) or the distribution of auROC values and 0.5 (Wilcoxon rank-sum test) are represented by the circles at the top. (**B**) Averaged firing rate for 5 s after the first licking in the HR trial (Kruskal-Wallis test, $\chi^2$ = 16.396, p = 2.8 × 10$^{-4}$, *post hoc* Tukey-Kramer test, early vs. middle, p = 0.726, early vs. late, p = 7.2 × 10$^{-4}$, middle vs. late, p = 0.009). (**C**) Averaged total number of time bins above 3 s.d. of the baseline firing for 5 s after the first licking in the HR trial (Kruskal-Wallis test, $\chi^2$ = 12.803, p = 0.002, *post hoc* Tukey-Kramer test, early vs. middle, p = 0.173, early vs. late, p = 0.001, middle vs. late, p = 0.219). (**D**) Cumulative probability of the proportion of first licking-related HR type neuron number (Kruskal-Wallis test, $\chi^2$ = 39.280, p = 3.0 × 10$^{-9}$, *post hoc* Tukey-Kramer test, early vs. middle, p = 1.3 × 10$^{-7}$, early vs. late, p = 0.995, middle vs. late, p = 2.3 × 10$^{-7}$). (**E**) Distribution of the correlation coefficient between the numbers of spikes and licking. Closed column indicates the number of neurons showing significant correlations between the two parameters (5 out of 107 neurons). (**F**) Firing rate (top rows), auROC (middle rows), and licking rate (bottom rows) after the reward of first licking-related HR neurons in the pVLS at the early (n = 17 neurons), middle (n = 11 neurons), and late (n = 64 neurons) stages. The data are aligned to the choice response, and the timings of the reward sound and first licking are shown. Time bins with significant differences between the rate in the HR trial and any of the rates in other trials or the distribution of auROC values and 0.5 are represented as above. (**G**) Averaged firing rate of pVLS neurons after the first licking in the HR trial (Kruskal-Wallis test, $\chi^2$ = 7.652, p = 0.022; *post hoc* Tukey-Kramer test, early vs. middle, p = 0.975, early vs. late, p = 0.076, and middle vs. late, p = 0.096). (**H**) Averaged total number of time bins above 3 s.d. of the baseline firing of pVLS neurons after the first licking in the HR trial (Kruskal-Wallis test, $\chi^2$ = 0.023, p = 0.989). (**I**) Cumulative probability of the proportion of first licking-related HR type neuron number (Kruskal-Wallis test, $\chi^2$ = 47.922, p = 3.9 × 10$^{-11}$; *post hoc* Tukey-Kramer test, early vs. middle, p = 0.011, early vs. late, p = 1.7 × 10$^{-4}$, and middle vs. late, p = 9.7 × 10$^{-10}$). (**J**) Distribution of the correlation coefficient between the number of spikes and the licking. Closed column indicates the number of neurons showing significant correlations between the two parameters (5 out of 92 neurons). Data are indicated as the mean ± s.e.m. (**A, F**) or the median and quartiles with the maximal and minimal values (**B, C, G, H**). **p < 0.01 and ***p < 0.001. Abbreviations: RS, reward sound; and FL, first licking.

The online version of this article includes the following source data and figure supplement(s) for figure 6:

**Source data 1.** Firing activity of first licking-related HR type neurons in the two striatal subregions.

**Figure supplement 1.** Sustained activity after the reward of first licking-related LL type neurons in the aDLS.

**Figure supplement 1—source data 1.** Firing activity of first licking-related LL type neurons in the aDLS.

*Figure 6 continued on next page*

*Figure 6 continued*

**Figure supplement 2.** Sustained activity after the reward of first licking-related LL type neurons in the pVLS.

**Figure supplement 2—source data 1.** Firing activity of first licking-related LL type neurons in the pVLS.

licking across the stages (*Figure 6—figure supplement 1A–C*). The proportion of first licking-LL type neurons during the learning processes displayed similar changes as observed in the first licking-HR neurons, and there were few neurons with significant correlations between the firing rate and licking performance (*Figure 6—figure supplement 1D, E*). These results indicate that the aDLS contained neurons showing long-lasting, increased activity after the reward in the specific correct combination of the stimulus and response (HR or LL trial), and that the extent of firing activity and length of firing period in the response appeared to be gradually reduced with no correlation between the proportion of neurons and the progress of the learning stages.

In the pVLS, the first licking-HR type neurons showed a sustained elevation in the firing rate after receiving a reward in the HR trial compared to that in the HL or LR trial throughout the three stages (*Figure 6F*, top rows). A similar elevation in the firing rate was also observed in the LL trial throughout the stages (*Figure 6F*, top rows). The auROC values revealed a significant difference of the HL or LR trial, but not the LL trial, from the HR trial throughout the stages (*Figure 6F*, middle rows). The average firing rate did not show significant changes among the three stages (*Figure 6G*), and the increased firing period was also similar across the stages (*Figure 6H*). The proportion of first licking-HR neurons increased at the late stage compared to the early or middle stage (*Figure 6I*). The firing rate showed almost no significant correlation to the licking performance (*Figure 6J*). In addition, the first licking-LL neurons exhibited similar firing patterns and changes in the proportion of neurons across the stages (*Figure 6—figure supplement 2A–E*). Therefore, the pVLS included neurons representing the long-lasting increased activity after the reward in the two kinds of correct responses (HR and LL trials) with the increased number of these neurons at the late stage of learning.

## Neurons in the pVLS, but not in the aDLS, exhibit firing patterns related to the beginning and ending of the behavior

The dorsal striatum plays a key role in the acquisition and execution of behaviors based on the stimulus-response association, and the association is accompanied by a transient activation of neuronal firing immediately after the stimulus presentation or action in trials, which is considered to represent the beginning and ending of a learned behavior (*Barnes et al., 2005*; *Jin and Costa, 2010*; *Jog et al., 1999*). We first examined whether the firing activity of cue onset-related HR and LL type neurons corresponds to the beginning of the behavior in the auditory discrimination task. In the aDLS, the firing rate of both types of neurons showed a significant difference at some time points in each trial compared to those in any of the other three trials, with a significant difference of auROC values between the trials (*Figure 7—figure supplement 1A, B*). However, the transiently increased activity did not emerge at the time points corresponding to the beginning of the behavior. In contrast, the cue onset-related HR type neurons in the pVLS showed a transient increase in the firing rate immediately after the cue onset, and the activity appeared at the middle and late stages (*Figure 7A*, top rows). The transient activation was observed not only in the HR trial but also in the other three trials, being consistent with the results of a previous report (*Barnes et al., 2011*). The increased firing rate showed a slight but notable difference between the HR and LL trials with a significant difference in the auROC values between the two kinds of trials (*Figure 7A*, bottom rows), suggesting that different combinations between the stimulus and response may affect the level of firing activity. The rate differences between the two trials were observed at the middle and late stages (*Figure 7B*). The proportion of cue onset-HR neurons increased at the middle and late stages compared to the early stage (*Figure 7C*). In addition, the firing rate of the cue onset-related LL type neurons exhibited patterns similar to those observed in the cue onset-HR neurons (*Figure 7—figure supplement 2A*). The data indicate that subpopulations of pVLS neurons show firing activity related to the beginning of the behavior, with a rising number of neurons throughout discrimination learning.

Next, we examined the firing activity of choice response-related HR and LL type neurons, corresponding to the ending of a behavior in auditory discrimination. In the aDLS, the firing rate of these neurons did not exhibit the transient increase related to the ending of the behavior. Instead, there was

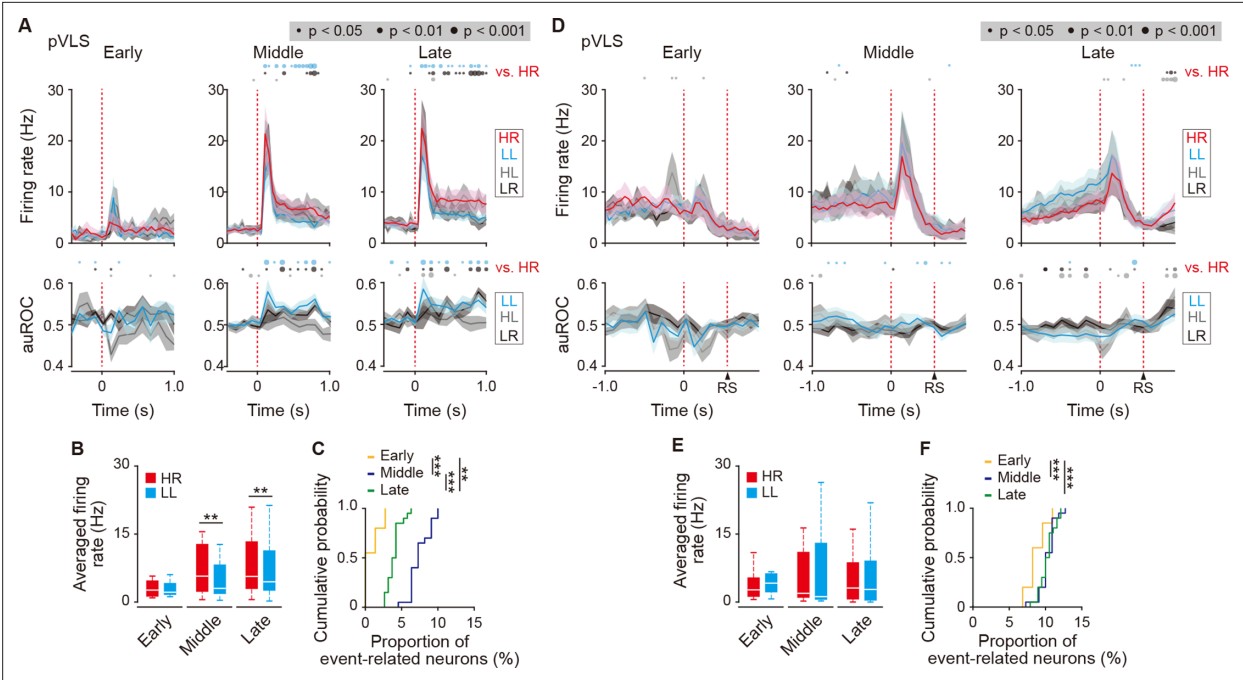

**Figure 7.** Transient activity related to the beginning and ending of a behavior of cue onset- and choice response-related HR type neurons in the pVLS. (**A**) Firing rate (top rows) and auROC values (bottom rows) of cue onset-related HR type neurons in the pVLS at the early (n = 4 neurons), middle (n = 21 neurons), and late (n = 24 neurons) stages. These data are aligned to the cue onset. Time bins with significant differences between the mean firing rate in the HR trial and either of the rate in the other three trials (Wilcoxon signed rank test) or the distribution of auROC values and 0.5 (Wilcoxon rank-sum test) are represented by the circles at the top. (**B**) Averaged firing rate of the cue onset-related HR type neurons in HR and LL trials (0–600ms after the cue onset; Wilcoxon signed rank test; early, p = 0.875, middle, p = 0.006, and late, p = 0.004). (**C**) Cumulative probability of the proportion of the cue onset-related HR type neuron number (Kruskal-Wallis test, $\chi^2$ = 51.114, p = 8.0 × 10$^{-12}$; *post hoc* Tukey-Kramer test, early vs. middle, p = 9.6 × 10$^{-10}$, early vs. late, p = 0.002, and middle vs. late, p = 6.4 × 10$^{-4}$). (**D**) Firing rate (top rows) and auROC values (bottom rows) of the choice response-related HR type neurons in the pVLS at the early (n = 11 neurons), middle (n = 22 neurons), and late (n = 41 neurons) stages. These data are aligned to the choice response. The timing of the reward sound is presented. Time bins with significant differences between the mean rate in the HR trial and any of the rates in other trials or the distribution of auROC values and 0.5 are represented as above. (**E**) Averaged firing rate of the choice response-related HR type neurons in HR and LL trials (0–600 ms after the choice response) (Wilcoxon signed rank test; early, p = 0.278, middle, p = 0.758, and late, p = 0.279). (**F**) Cumulative probability of the proportion of the choice response-related HR type neuron number (Kruskal-Wallis test, $\chi^2$ = 13.786, p = 0.001; *post hoc* Tukey-Kramer test, early vs. middle, p = 0.003, early vs. late, p = 0.005, and middle vs. late, p = 0.976). Data are indicated as the mean ± s.e.m. (**A, D**) or the median and quartiles with the maximal and minimal values (**B, E**). **p < 0.01 and ***p < 0.001. Abbreviation: RS, reward sound.

The online version of this article includes the following source data and figure supplement(s) for figure 7:

**Source data 1.** Firing activity of cue onset- and choice response-related HR type neurons in the pVLS.

**Figure supplement 1.** Firing activity of cue onset-related HR and LL type neurons in the aDLS.

**Figure supplement 1—source data 1.** Firing activity of cue onset-related neurons in the aDLS.

**Figure supplement 2.** Firing activity of cue onset- and choice response-related LL type neurons in the pVLS.

**Figure supplement 2—source data 1.** Firing activity of cue onset- and choice response-related LL type neurons in the pVLS.

**Figure supplement 3.** Firing activity of choice response-related HR and LL type neurons in the aDLS.

**Figure supplement 3—source data 1.** Firing activity of choice response-related neurons in the aDLS.

a modest increase before the lever press (*Figure 7—figure supplement 3A, B*). In contrast, the choice response-related HR type neurons in the pVLS were transiently activated immediately after the lever press in the four kinds of trial at the middle and late stages, and the rate appeared to gradually decline toward the reward sound presentation (*Figure 7D*). The averaged firing rate was similar between the HR and LL trials through the stages (*Figure 7E*). The proportion of the choice response-HR neurons increased at the middle and late stages compared to the early stage (*Figure 7F*). In addition, the rate of choice response-related LL type neurons displayed similar firing patterns to those observed in the choice response-HR neurons (*Figure 7—figure supplement 2B*). Thus, subpopulations of pVLS neurons also represent the activity related to the ending of the behavior in auditory discrimination,

increasing the neuron number throughout the learning stages in a similar pattern to neurons relevant to the beginning of the behavior in the task.

## Structure of neural circuits linked to the aDLS and pVLS

Anatomically, striatal subregions are known to form heterogeneous cortico-basal ganglia loops in mice (*Foster et al., 2021*; *Hintiryan et al., 2016*; *Hunnicutt et al., 2016*). To characterize the structure of neural circuits connected to the aDLS and pVLS in rats, a solution containing cholera toxin subunit B (CTb) labeled with Alexa 488 or Alexa 555 (1.0 mg/mL, 0.3 μL/site) as a bidirectional tracer (*Amita et al., 2019*) was injected into the aDLS and pVLS of the same animals, respectively (*Figure 8A*). Brain sections were prepared, and cell bodies and axon terminals were detected by CTb with fluorescence signals. Sections through the aDLS and pVLS indicated the fluorescence signals around the injection sites (*Figure 8B*). Alexa 488 signals were localized in cell bodies in the bilateral, rostrodorsal regions of some cortical areas, including the primary motor cortex, secondary motor cortex, primary somato-sensory cortex, and secondary somatosensory cortex, and in the ipsilateral regions of various thalamic nuclei, such as the centrolateral nucleus, centromedian nucleus, parafascicular nucleus, and ventrome-dial nucleus, whereas Alexa 555 signals were observed in the bilateral, ventrolateral regions of some cortical areas, including the most rostral parts of the primary motor cortex, agranular/dysgranular/granular insular cortices, rhinal cortex, and amygdala nucleus, as well as in the ipsilateral regions of various thalamic nuclei, such as the centromedian nucleus, parafascicular nucleus, and medial genicu-late nucleus (*Figure 8C and D*). In addition, Alexa 488 signals were distributed in axon terminals in the anterior part of the globus pallidus (GP) and entopeduncular nucleus (EPN), whereas Alexa 555 signals were detected in the posterior GP (*Figure 8E*). In the ventral midbrain, Alexa 488 and 555 signals in the nerve terminals were distributed distinctively in the ventromedial and dorsolateral parts of the substantia nigra pars reticulata (SNr), respectively, along with the rostrocaudal axis (*Figure 8F*). In the cell bodies, both signals were localized in the centromedial and lateral parts of the substantia nigra pars compacta (SNc), respectively (*Figure 8F*). The signals in the SNc were colocalized with immuno-reactivities for TH, a marker of dopamine neurons in the ventral midbrain (*Figure 8G*). These results indicate that the aDLS and pVLS receive innervations from different subregions in the cerebral cortex, intralaminar thalamus, and SNc, and send projections to different subregions in the basal ganglia nuclei including the GP, EPN, and SNr, being consistent with previous anatomical studies of cortico-basal ganglia loops (*Foster et al., 2021*; *Hintiryan et al., 2016*; *Hunnicutt et al., 2016*).

## Discussion

Our findings demonstrate that the aDLS and pVLS are required for the acquisition of auditory discrim-ination at different stages of learning, and that the DMS is not necessary for the acquisition of discrim-ination, although it appears to contribute to the execution of actions, in particular to the regulation of response time. These results not only support some recent studies reporting no requirement of the DMS in procedural learning (*Featherstone and McDonald, 2004*; *Smith et al., 2021*; *Wolff et al., 2022*), but also reveal important roles of two different subregions in the lateral striatum in external cue-dependent decision-making. We propose that aDLS and pVLS neurons act to integrate the new learning of auditory discrimination in spatiotemporally and functionally distinct manners, showing a remarkable difference from the prior learning model that proposes the functional dominance changes from the DMS to DLS subregions during procedural learning (*Dickinson, 1985*; *O'Hare et al., 2016*; *Redgrave et al., 2010*; *Yin et al., 2004*; *Yin et al., 2005*).

Our $^{18}$F-FDG-PET imaging study demonstrated that the brain activity in the aDLS reaches a peak at the middle stage of the acquisition phase of auditory discrimination (*Figure 2J*). The results of the excitotoxic lesion experiment of striatal subregions suggest that the aDLS functions to enhance the acquisition of discrimination through the promotion of behavior based on the stimulus-response association and the suppression of behavior based on the response-outcome association (*Figure 3G and J*). Transient inhibition experiment of neuronal activity confirmed this aDLS function at the middle stage of the learning, although the drug inhibition tended to reduce the stimulus-response behavior (*Figure 4C, E and F*). In addition, we found subpopulations of aDLS neurons (reward sound-related neurons) which represent the behavioral outcome and differentiate the correct (HR or LL) and incor-rect (HL or LR) choices (*Figure 5D*; *Figure 5—figure supplement 4A*). The distinct responses of

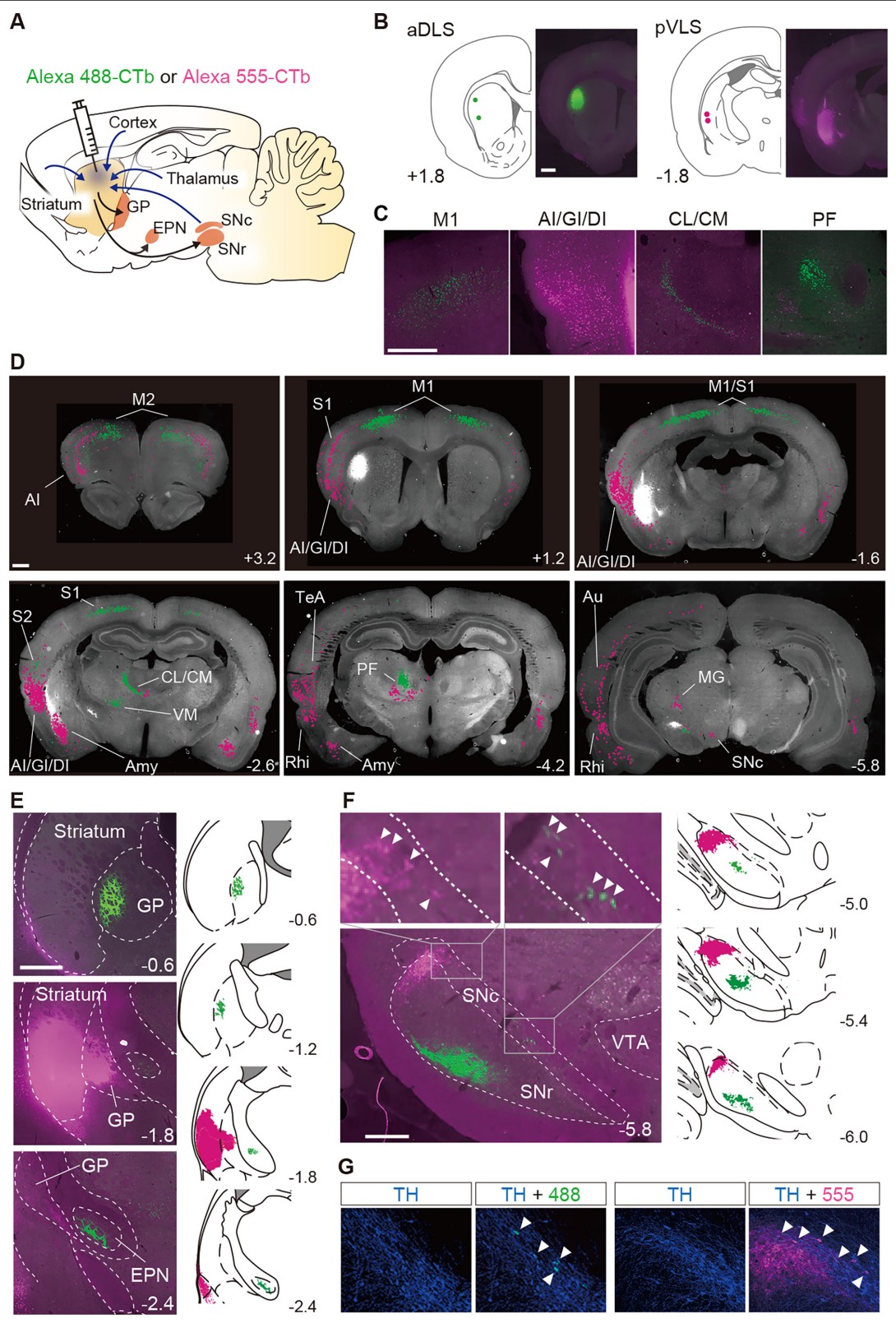

**Figure 8.** Segregation of neural circuits linked to the aDLS and pVLS. (**A**) Schematic illustration of the major inputs to and outputs from the striatum. A solution containing CTb labeled with Alexa 488 (green) or Alexa 555 (magenta) was injected into the aDLS and pVLS of the same animals (n = 2). (**B**) Coordinates for tracer injection into the aDLS and pVLS. Right, representative fluorescence images of sections through the injection sites. (**C**) Typical images of sections through the cerebral cortex and intralaminar thalamus. (**D**) Distribution of cell bodies innervating the aDLS and pVLS. Fluorescence

*Figure 8 continued on next page*

*Figure 8 continued*

signals in the cell bodies were marked by green or magenta dots using ImageJ (version 1.53) and overlaid on gray-scaled images of a brain section. (**E**) Axonal terminals arising from the aDLS and pVLS to the GP or EPN. Representative images of tracing are shown in the left panels. Camera lucida drawings of labeled signals are shown in the right panels. (**F**) Labeling of axonal terminals and cell bodies in the ventral midbrain. Representative images are shown in the left panels. Upper images are magnified views of the rectangles in the lower image. Camera lucida drawings are shown in the right panels. Arrowheads in the magnified images indicate labeled cell bodies. (**G**) Triple-fluorescence histochemistry with sections through the ventral midbrain. The sections were stained with an anti-TH antibody and then with a secondary IgG conjugated with Alexa 647. Representative images of the centromedial (left) and lateral (right) parts of the SNc are shown. Arrowheads indicate TH-positive cells with CTb-labeled signals in each image. The anteroposterior coordinates from bregma (mm) are shown. Scale bars; 1 mm (**B–D**), 500 µm (**E**), 100 µm (**F, G**). Abbreviations: AI, agranular insular cortex; Amy, amygdala nucleus; Au, auditory cortex; CL, centrolateral nucleus; CM, centromedian nucleus; DI, dysgranular insular cortex; GI, granular insular cortex; M1, primary motor cortex; M2, secondary motor cortex; MG, medial geniculate nucleus; PF, parafascicular nucleus; Rhi, rhinal cortex; S1, primary somatosensory cortex; S2, secondary somatosensory cortex; TeA, temporal association cortex; VM, ventromedial nucleus; and VTA, ventral tegmental area.

these neurons may explain the mechanism by which the aDLS regulates the stimulus-response and response-outcome associations through distinct neural circuits leading to a differential control of the two associations. Alternatively, the responses mainly promote the behavior based on the stimulus-response association, thereby secondarily inhibiting the response-outcome association. Indeed, some previous studies have suggested that an action based on either the stimulus-response association or response-outcome association is generated as a result of competition between different striatal subregions (*Gremel et al., 2016*). We also found other subpopulations of aDLS neurons (first licking-related neurons) showing long-lasting increased activity after obtaining reward in response to a specific combination between the stimulus and response (HR or LL trial; *Figure 6A*; *Figure 6—figure supplement 1A*). This activity may function as the feedback signal to promote the stimulus and response association, resulting in the enhancement of discrimination learning. The proportion of neurons related to the behavioral outcome showed a gradual reduction through the progress of auditory discrimination (*Figure 5F*; *Figure 5—figure supplement 4C*), and the extent of the firing activity and the length of the firing period of neurons showing sustained activation after reward were also decreased at the late stage of learning (*Figure 6B and C*). The learning-dependent changes in the properties of these subpopulations in the aDLS may explain the mechanism underlying the earlier acquisition process of auditory discrimination.

The $^{18}$F-FDG-PET imaging indicated that the pVLS activity was gradually elevated through the acquisition phase of auditory discrimination, showing the maximal value at the late stage (*Figure 2K*). The lesion experiment suggested that the pVLS drives the formation of discrimination learning via the selective progression of the stimulus-response association (*Figure 3H and K*), and the transient inhibition experiment ascertained the pVLS function at the late stage of learning (*Figure 4D, G and H*). Previous studies have shown that the stimulus-response association is encoded in the DLS or VLS as the transient neuronal activity representing the beginning and ending of a learned behavior (*Barnes et al., 2005*; *Chen et al., 2021*; *Jin and Costa, 2010*; *Jog et al., 1999*). A recent study reported that the VLS is causally related to learned performance based on the auditory stimulus, while the DLS shows a weaker relationship to the behavior (*Lee et al., 2020*). In the present study, we found subpopulations of pVLS neurons that represent the beginning and ending of a behavior (cue onset- and choice response-related neurons, respectively; *Figure 7A and D*; *Figure 7—figure supplement 2A, B*), showing an increased number of neurons along with the progress of discrimination learning (*Figure 7C and F*). These data suggest that pVLS neurons contribute to the formation of the stimulus-response association through the learning processes. In addition, we found another subpopulation of pVLS neurons (first licking-related neurons) that are continuously activated after the reward outcome in response to two kinds of correct choices (HR and LL trials; *Figure 6F*; *Figure 6—figure supplement 2A*), and the proportion of neurons was increased at the late stage (*Figure 6I*; *Figure 6—figure supplement 2D*). This activity may act as a feedback signal to consolidate the stimulus and response association, leading to the maintenance of associative memory at the late stage during the acquisition phase of learning.

In the present study, we identified neurons with four kinds of event-related activity in the aDLS and pVLS and classified them into the HR and LL type neurons (*Figures 5–7*; *Figure 5—figure supplements 4 and 5*; *Figure 6—figure supplements 1 and 2*; *Figure 7—figure supplements 1–3*). Parts of these two types of neurons with the respective event-related activity overlapped, and

the numbers of overlapping neurons varied between striatal subregions. In the pVLS, the overlapping neurons for the same event-related activities were 35.1%, 45.2%, and 67.0% in the cue onset, choice response, and first licking, respectively. In contrast, the aDLS had lower overlaps of 12.6% and 22.7% in the reward sound and first licking, respectively. Thus, distinct neurons in the aDLS tend to have properties of a single trial type, while the same neurons in the pVLS seem to possess properties of both types. One possible explanation for the diversity in the proportion of overlapping neurons between the two subregions may be the difference in input patterns into striatal neurons or plastic changes in functional connectivity during learning processes. The detailed mechanism by which aDLS and pVLS neurons react to the events in a different manner should be investigated in the future.

We examined the neural circuits linked to the aDLS and pVLS in rats by using bidirectional tracers. The results showed that these two striatal subregions receive innervations from different subregions in the cerebral cortex, intralaminar thalamus, and ventral midbrain and provide projections to different subregions in the basal ganglia nuclei (*Figure 8*), confirming that the two subregions form the heterogeneous cortico-basal ganglia loops in the rodent brain (*Foster et al., 2021*; *Hintiryan et al., 2016*; *Hunnicutt et al., 2016*). These data suggest that the aDLS and pVLS mediate the acquisition of auditory discrimination via the parallel loops and cooperate to achieve neural processing related to the dual regulation of the stimulus-response and response-outcome associations. Further studies are needed to elucidate the mechanism by which the two kinds of loops are processed separately and integrate the new learning of discrimination.

## Limitations

In the $^{18}$F-FDG PET neuroimaging study, the brain activity in the aDLS reached a peak at the middle stage of the acquisition phase of auditory discrimination (*Figure 2J*). In the multi-unit electrophysiological recording experiment, the firing activity of aDLS neuron subpopulations related to the behavioral outcome showed no significant differences among the three stages (*Figure 5E*; *Figure 5—figure supplement 4B*), while the proportions of these subpopulations gradually reduced through the progress of learning stages (*Figure 5F*; *Figure 5—figure supplement 4C*). The extent of firing activity and length of firing period of other subpopulations showing sustained activation after the reward appeared to show learning-dependent decrease (*Figure 6B and C*; *Figure 6—figure supplement 1B, C*), although the proportions of these subpopulations indicated no correlation to the progress of the learning (*Figure 6D*; *Figure 6—figure supplement 1D*). The patterns of the temporal changes in the brain activity of the striatal subregions during the learning stages did not completely match the time variation in the property and proportion of specific event-related neurons. In our electrophysiological analysis, we identified well-isolated neurons from striatal subregions across the behavioral task, focusing on putative MSNs (*Figure 5—figure supplement 2*). Based on the combinatorial pattern of the tone instruction cue and lever press, we categorized the electrophysiological data into the four trials, including the HR, LL, HL, and LR. We identified HR or LL type neurons showing significant changes in the firing rate related to specific events compared to the baseline firing rate. These neurons were further divided into two groups with increased or decreased activity relative to the baseline firing (*Figure 5—figure supplement 3*). In the present study, we focused on event-related neurons with increased activity. Because the analysis was limited to neuronal subpopulations related to specific events with the increased activity, we consider that it is difficult to explain fully dynamic shifts in the brain activity of the striatal subregions dependent on the progress of learning by the time variation of firing activity of individual event-related neurons. The activity of other subpopulations in the striatum may be involved in the shifts in brain activity during the learning processes. In addition, recent studies have reported that the activity of glial cells influences the uptake of $^{18}$F-FDG and that these cells regulate spike timing-dependent plasticity (*Valtcheva and Venance, 2016*). Changes in glial cellular activity, through the control of synaptic plasticity, may partly contribute to the pattern formation of learning-dependent shift of brain activity.

# Materials and methods

## Key resources table

| Reagent type (species) or resource | Designation | Source or reference | Identifiers | Additional information |
|---|---|---|---|---|
| Strain, strain background (*Rattus norvegicus*, Long-Evans) | Long-Evans Rats | Institute for Animal Reproduction | | Wild-type rats |
| Antibody | Donkey anti-mouse IgG antibody conjugated with Alexa Fluor 647 (polyclonal) | Thermo Fisher Scientific | Cat# A-31571; RRID:AB_162542 | 1:1,000 |
| Antibody | Goat anti-mouse IgG antibody conjugated with Alexa Fluor 488 (polyclonal) | JacksonImmunoresearch Laboratories | Cat# 715-545-151; RRID:AB_2341099 | 1:1,000 |
| Antibody | Mouse anti-NeuN antibody (monoclonal) | Millipore | Cat# MAB377; RRID:AB_2298772 | 1:1,000 |
| Antibody | Mouse anti-TH antibody (monoclonal) | Millipore | Cat# MAB318-AF488; RRID:AB_2201528 | 1:1,000 |
| Chemical compound, drug | (±)-Ibotenic acid | FUJIFILM Wako Pure Chemical | Cat# 533–54971 | 0.8 mg/mL |
| Chemical compound, drug | Muscimol | Sigma-Aldrich | Cat# M1523 | 0.1 µg/µL |
| Chemical compound, drug | Saccharin sodium salt dihydrate | Tokyo Chemical Industry | Cat# B0131 | 0.02% |
| Software, algorithm | Kilosort2 | Marius Pachitariu; *Stringer et al., 2019* | RRID:SCR_016422 | https://www.github.com/MouseLand/Kilosort2 |
| Software, algorithm | KlustaKwik2 | *Kadir et al., 2014*; *Goodman et al., 2015* | RRID:SCR_014480 | https://github.com/kwikteam/klustakwik2 |
| Software, algorithm | MATLAB | MathWorks | RRID:SCR_001622 | |
| Software, algorithm | phy | cortex-lab; *Rossant et al., 2024* | RRID:SCR_016249 | https://github.com/cortex-lab/phy |
| Software, algorithm | PMOD version 3.2 | PMOD Technologies | RRID:SCR_016547 | https://www.pmod.com/web/ |
| Software, algorithm | SPM version 8 | Wellcome Department of Imaging Neuroscience | RRID:SCR_007037 | https://www.fil.ion.ucl.ac.uk/spm/ |
| Software, algorithm | SPSS version 27 | IBM | RRID:SCR_002865 | |
| Other | Cholera toxin subunit B conjugated to Alexa 488 | Invitrogen | Cat# C34775; RRID:AB_2536189 | 1.0 mg/mL |
| Other | Cholera toxin subunit B conjugated to Alexa 555 | Invitrogen | Cat# C34776; RRID:AB_968419 | 1.0 mg/mL |
| Other | Cresyl violet | Muto Pure Chemicals | Cat# 41021 | 0.1% |
| Other | DAPI | Thermo Fisher Scientific | Cat# D1306; RRID:AB_2629482 | 0.1 µg/mL |
| Other | DAPI and red fluorescent Nissl stain solution | Thermo Fisher Scientific | Cat# N21482; RRID:AB_2620170 | 1:10,000/ 1:200 |
| Other | Grain-based precision pellets | Bio-serv | Cat# F0021 | 45 mg |
| Other | Guide cannula | P1 Technologies | Cat# C315I | 26 gauge |
| Other | Internal cannula | P1 Technologies | Cat# C315G | 30 gauge |
| Other | Laboratory chow | CLEA Japan | Cat# CE-2 | |
| Other | Silicon probe | NeuroNexus | Buzsaki64spL | 64-channel (6-shank) |
| Other | Templates for the Long-Evans rat brain | Neuroimaging Tools and Resources Collaboratory | RRID:SCR_003430 | https://www.nitrc.org/projects/tpm_rat/ |
| Other | Thin wall borosilicate glass | Sutter Instrument | Cat# B100-75-10 | 60 µm diameter |

## Animals

All procedures for animal care and handling were approved by the Institutional Animal Care and Use Committees of RIKEN Center for Biosystems Dynamics Research, Fukushima Medical University, and

Osaka City University. Animal procedures were carried out in accordance with the guidelines established by their Institutional Committees. All efforts were made to minimize the number of animals used and their suffering throughout the course of the experiments. Male Long-Evans rats (8–13 weeks old, Institute for Animal Reproduction, Ibaraki, Japan) were used for the present study. The rats were kept at 22 ± 2 °C and 60% humidity in a 12 hr light/12 hr dark cycle and were continuously available for laboratory chow (CE-2, CLEA Japan, Fukuoka, Japan) and water. Three days before the behavioral training, the rats were subjected to dietary restriction to maintain their body weight at ~85% of the free feeding weight or water restriction in which they received free access to water for 6–9 hr each day.

## Surgery

Rats underwent a stereotaxic surgery under isoflurane anesthesia (1–5%). For excitotoxic lesion of striatal neurons, a solution containing IBO (533–54971, FUJIFILM Wako Pure Chemical, Osaka, Japan) in PBS (0.8 mg/mL, 0.3 µL per site) was injected bilaterally with the coordinates (mm) from bregma and dura for the aDLS (AP +1.6, ML ±3.3, DV –3.9), pVLS (AP –1.6, ML ±4.6, and DV –5.6), or DMS (AP –1.6, ML ±1.4, and DV –4.8) according to the atlas of rat brain (*Paxinos and Watson, 1998*). Injection was performed through a thin-wall borosilicate glass pipette (diameter: 60 µm, B100-75-10, Sutter Instrument, Novato, CA) at a constant velocity of 0.1 mL/min with the microinfusion pump (ESP-32, EICOM, Kyoto, Japan). For transient inactivation of striatal neurons, two guide cannulae (length: 10 mm, outer diameter: 26 gauge, C315G, P1 Technologies, Roanoke, VA) were bilaterally placed on the skull and fixed using four screws and dental cement. The chip of the cannula was placed at 1 mm above the target coordinates. A solution containing MUS (M1523, Sigma-Aldrich, St. Louis, MO) in saline (0.1 µg/µL, 0.25 µL per site) was injected into the target coordinates through the internal cannula (outer diameter: 30 gauge, C315I, P1 Technologies) at a constant velocity of 0.25 µL/min. For the electrophysiological experiments, surgery was performed as described previously (*Kitanishi et al., 2021*). A 64-channel silicon probe (Buzsaki64spl, NeuroNexus, Ann Arbor, MI) attaching to a 3D-printed microdrive with a movable screw was chronically implanted with the coordinates (mm) from the bregma and dura into the left aDLS (AP +1.6, ML –3.3, and DV –2.5) and pVLS (AP –1.6, ML –4.6, and DV –3.6). A Faraday cage composed of copper mesh was secured on the skull with dental cement to reduce electrical noise and avoid damage to the implants. For anterograde and retrograde tracing, a solution containing CTb conjugated to Alexa 488 (1.0 mg/mL, C34775, Invitrogen, Waltham, MA) or CTb conjugated to Alexa 555 (1.0 mg/mL, C34776, Invitrogen) was injected into the aDLS and pVLS (0.3 µL/site), respectively. The coordinates for the injection into striatal subregions were the same as described above. The rats were allowed to recover for 1 week after the surgeries.

## Behavioral analysis

The behavioral tasks were conducted in operant conditioning chambers (30.5 × 24.1 × 29.2 cm; ENV-007, MED Associates, Fairfax, VT) equipped with two retractable levers on both sides of a reward port in the front panel (ENV-203M-45, ENV-112CM, and ENV-201A, MED Associates). A multi-tone generator (ENV-223, MED Associates) and room light were mounted on the top of the center and corner panels on the rear side, respectively. For the electrophysiological experiment, a 3D printed reward port was used to detect the licking and to prevent contact with the head attachment and front panel. The behavioral experiment was performed using reinforcement pellets (grain-based precision pellets; F0021, Bio-serv, Flemington, NJ) as a food reward. To prevent spillover of radioactivity into the temporal cortex as a consequence of excessive $^{18}$F-FDG accumulation into the muscles by their activity to eat the pellets (*Endepols et al., 2010*; *Marx et al., 2012*), the PET imaging experiment was carried out using 0.02% saccharin solution (B0131, Tokyo Chemical Industry, Tokyo, Japan) as a liquid reward. The electrophysiological experiment was also carried out by using the saccharin solution.

The training schedule consisted of four continuous steps. In the first step (magazine training), rats were habituated to the operant conditioning chambers, and the reward was presented with the click every 20 s. Each daily session was started by illumination of the room light and continued for 30 min. The sessions were conducted for 3 consecutive days. In the second step (shaping), only one lever on the left or right side was inserted. When the rats pressed the lever, they earned a reward accompanied by a click. Each daily session was started by the illumination of the light and lasted 30 min or until the criteria, in which the rats received 90 rewards. When the number of lever presses on either

side reached the criteria, the lever was inserted in the opposite side on the next day. When the rats achieved the criteria on both sides for 2 or 3 days, they moved to the following step. In the third step (single lever press task), each trial was started by illumination of the light and insertion of one lever on either the left or right side in a pseudorandom manner with equal frequencies on both sides for every set of four trials. When the rats pressed the lever, the reward was presented with the sound, and the lever was immediately retracted. The room light turned off 4 s after the lever press. When they did not press the lever within 10 s after the start, the trial was terminated when the light turned off. The trial was repeated every 20 s. Each daily session continued for 30 min, and the sessions were conducted for 4 consecutive days. The number of lever presses in each session was counted. In the final step (two-alternative auditory discrimination task), each trial was started by the presentation of a high (10 kHz) or low (2 kHz) frequency tone as an instruction cue in a pseudorandom manner. After 3 s, the room light was illuminated, and two retractable levers were inserted at the same time. The rats were required to press the left or right levers in response to the low or high tone cues, respectively. When the rats pressed the correct lever, the reward was presented with the sound, and the levers were immediately retracted. Correct response allowed the rats to access the reward: the retractable spout delivering water was inserted for 1 s for the PET imaging experiment; a pellet was delivered by a food dispenser for the pharmacological experiment; and three drops of water were delivered through a solenoid valve for the electrophysiological experiment. The room light turned off 4 s after the correct response. On the other hand, the error response resulted in the retraction of levers and the turning off of the room light. No response within 10 s after the lever insertion was recorded as an omission response. The success rate in each session was calculated by dividing the number of correct response trials by the number of lever press trials minus the number of omission trials. The trial was repeated every 20 s. Each daily session continued for 60 min, and the sessions were conducted for 14 or 24 days according to the experiments.

To assess the behavioral strategies in the auditory discrimination task, a trial-by-trial analysis was conducted; the win-shift-win strategy was defined as a correct response, in which, after a correct response during the previous trial, the rats press the opposite lever during the current trial in response to the shift of the tone instruction cue; the lose-shift-lose strategy was defined as an error response, in which, after an error response during the previous trial, the rats press the opposite lever during the current trial despite the shift of the instruction cue. The proportion of both strategies was calculated by dividing the number of each strategy by the number of lever press trials minus the number of omission trials.

For the transient inactivation experiment, MUS injections were bilaterally performed 20 min before the start of the session. For habituation, all rats were injected with saline bilaterally into the corresponding striatal subregions on the last day (Day 4) of the single lever press task. During the auditory discrimination task, the first injection was conducted on Day 2 of the task. The second and third injections were conducted the next day, after the success rate had exceeded 60% and 80%, respectively. Each session continued for 15 min.

For the electrophysiological experiments, a delay period (0.5 ± 0.2 s) was added in the auditory discrimination task between the lever press and reward sound to distinguish the motor- and reward-related responses of neurons. When the rats pressed the lever, it was immediately retracted. After the delay period, the reward was presented with a sound. The room light turned off 4 s after the lever press. The trial was repeated every 20 s, and each daily session continued for 60 min.

## PET imaging analysis

An [18]F-FDG-PET scan was performed with a microPET Focus220 (Siemens Medical Solutions, Knoxville, TN) designed for high-resolution imaging of small animals (spatial resolution of 1.4 mm in full width at half maximum at the center of the field of view) as described previously (*Cui et al., 2015*). Serial [18]F-FDG-PET scans were performed on the last day of the single lever press task and on Days 2, 6, 10, and 24 of the auditory discrimination task. To administer [18]F-FDG under the awake condition, rats were anesthetized through an indwelling catheter attached to their tail with 5% induction and 2.0–2.5% maintenance of isoflurane (median [interquartile range]: 6 [5-8] min). The recovery period was set to at least 2 hr until the beginning of the behavioral session, with the aim of minimizing the impact of anesthesia as much as possible. A solution of [18]F-FDG (ca. 74 MBq/0.4 mL) was intravenously injected just before the beginning of each behavioral session through the indwelling catheter. After

the 30 min session, the rats were returned to their home cage, and then a 30 min static PET scan was performed under anesthesia with 2.0–2.5% isoflurane 55 min after the $^{18}$F-FDG injection. During the PET scan, the body temperature of the anesthetized rats was maintained at 37 °C using a small animal warmer (BWT-100A, Bio Research Center, Nagoya, Japan). The acquired emission data were sorted into a single sinogram and reconstructed by a standard 2D filtered back projection (FBP) with a ramp filter and cutoff frequency at 0.5 cycles per pixel or by a statistical maximum a posteriori probability algorithm (MAP), 12 iterations with point spread function effect.

Voxel-based analysis of PET images was conducted according to the method described previously (*Cui et al., 2015*). Briefly, individual MAP-reconstructed FDG images were aligned to an FDG template image of Long-Evans rats using the PMOD software package (version 3.2; PMOD Technologies, Fällanden, Switzerland; https://www.pmod.com/web/) and then the aligned images were transformed into the space of T2 weighted MRI templates for the Long-Evans rat brain (Neuroimaging Tools and Resources Collaboratory; https://www.nitrc.org/projects/tpm_rat/). The transformation parameters obtained from individual MAP-reconstructed FDG images were used to match the individual FBP-reconstructed FDG images with the MRI templates. The voxel size of the template was magnified by 10 to a size similar to that of a human brain. All FBP-reconstructed FDG images were resampled with a voxel size of 1.2 × 1.2 × 1.2 mm and spatially smoothed through an isotropic Gaussian kernel (6 mm FWHM). Voxel-based statistical analysis was performed by using the Statistical Parametric Mapping (SPM) software (version 8; Wellcome Department of Imaging Neuroscience, London, UK; https://www.fil.ion.ucl.ac.uk/spm/). For visualization of the results, SPM t-statistic values were overlaid on MRI templates to define the voxels showing significantly increased or decreased brain activity, using a high threshold (p=0.001, uncorrected) and an extent threshold (with the output values from the voxel-based analysis). The voxel of interest was determined as voxels with more than 85% of the maximum T-value within activated areas obtained by voxel-based statistical comparison of the FBP-FDG images. The average regional $^{18}$F-FDG uptake was calculated on the basis of values of the voxel of interest in the images.

## Electrophysiological recording

Extracellular multi-unit recording was conducted at a sampling rate of 20 kHz from the striatum of freely moving rats by using a 512-ch acquisition board (Open Ephys, Atlanta, GA) via 64-ch recording headstages (C3325, Intan Technologies, Los Angeles, CA). The tips of the probe were lowered by turning the screw (> 140 μm/day) until it reached the target regions (the aDLS and pVLS) during the single lever press task. The tips were sometimes lowered by the same procedure (140–280 μm/day) after the daily recording session to improve cell yield. Spike sorting was performed automatically by the Kilosort2 software (*Stringer et al., 2019*; Marius Pachitariu, Ashburn, VA; *Pachitariu et al., 2024*). Clear noise clusters were removed in the first manual sorting using the phy software (*Rossant et al., 2024*). The remaining clusters were then re-clustered by using the KlustaKwik2 software (*Kadir et al., 2014*; *Goodman et al., 2015*), and the clusters were adjusted in the second manual sorting by using the phy again. The sorted clusters satisfying all the following criteria were used for further analysis: L-ratio < 0.05, isolation distance >15, and inter-spike interval (ISI) index < 0.2 (*Fee et al., 1996*; *Schmitzer-Torbert et al., 2005*). Well-isolated single neurons (n = 1485; aDLS, n = 1062; pVLS, n = 423) were classified as putative MSNs, fast spiking interneurons (FSIs), tonically active neurons (TANs), and unclassified interneurons (UIs). Putative MSNs and TANs were separated from FSIs and UIs by trough-to-peak duration of mean spike waveform (> 0.6 ms) according to previous studies with slight modifications (*Bakhurin et al., 2016*; *Kim et al., 2019*). Putative FSIs were then separated from UIs by utilizing the proportion of time associated with long ISIs. This was achieved by summing the ISIs longer than 2 s and subsequently dividing the resulting sum by the total recording time (Prop$_{ISIs}$ > 2 s; *Peters et al., 2021*). Neurons with values of Prop$_{ISIs}$ > 2 s less than 0.4 were classified as putative FSIs, while neurons with Prop$_{ISIs}$ > 2 s greater than 0.4 were classified as putative UIs. The putative MSNs and TANs were separated by measuring the post-spike suppression (*Peters et al., 2021*). We measured the length of suppressed time during which the firing rate of a neuron was suppressed following a spike by counting the number of 1-ms bins in its autocorrelation function until the rate of the neuron was equal to or greater than its averaged rate over the 600-ms to 900-ms autocorrelation bins. Putative TANs were separated by post-spike suppression of less than 40 ms, and the remaining neurons were classified as putative MSNs.

To divide the three stages during the acquisition phase, the sigmoidal function was fitted using the following four-parameter logistic equation:

$$f_{(x)} = \frac{1 - \gamma - (1 - \lambda)}{1 + \exp(-\beta(x - \alpha))} + \gamma$$

where α is the inflection point, β is a variable for slope, and $\lambda$ and γ are parameters related to the lower asymptote and upper asymptote of the learning curve, respectively. The function was used to evaluate the day when the success rate reached 20% ($t_{20\%}$) or 90% ($t_{90\%}$) of the maximum rate change defined by the difference between the γ and $\lambda$. Three stages were defined according to the following criteria: early stage (up to 3 days before the middle stage); middle stage (3 days including before and after the day of $t_{20\%}$); and late stage (3 days including the day of $t_{90\%}$ and before). One rat learned the discrimination faster, and the success rate reached $t_{90\%}$ 2 days after the day of $t_{20\%}$. One day before the day of $t_{90\%}$ was assigned to the late stage. For the following analysis, spike data in each striatal subregion were separated into the early, middle, and late stages (n=295, 364, and 360, respectively, for the aDLS; and n=87, 110, and 190, respectively, for the pVLS).

To determine the neurons showing significant changes in firing rate related to the task events, electrophysiological data were grouped into four trial types based on the instruction cue (high or low tone) and the choice (right or left lever); HR, high tone and right lever press; LL, low tone and left lever press; HL, high tone and left lever press; and LR, low tone and right lever press. Neurons with three or more trial data were analyzed. We identified the HR or LL type neurons, detecting significant increases or decreases of the firing rate by applying the Wilcoxon rank-sum test (p<0.01) to trial-by-trial spike number between the baseline period (BL, –1500 to –500 ms or –1100 to –500 ms before presentation of the instruction cue) and any of four peri-event periods (cue onset, 0 to +1000 ms after the onset of the instruction cue; choice response, –700 to +300 ms around the time of lever press; reward sound, –300 to +300 ms around the time of sound onset instructing the reward delivery; and first licking, 0 to +1000 ms after the time of first licking after the choice response). Any spikes around the events were averaged in 50-ms bins. The event-related activity was shown as averaged firing rate aligned based on the timing of an event for each trial. The auROC values were calculated by using the spike counts between one trial and either of the other three trials. Bins showing significant differences in the auROC data were determined by applying the Wilcoxon rank-sum test (p<0.05) to the distribution of 200-ms bin-by-bin auROC values and 0.5.

To compare the number of neurons showing event-related activity among different stages, the number of HR or LL type neurons was calculated by matching the trial number at each stage. Twenty trials were randomly selected for a neuron, and neurons with less than 20 trials were removed from the analysis. The neurons with event-related activity were identified as described above. The proportion (percentage) of neurons relative to the total number of neurons was calculated at each individual stage. The same analysis was repeated 30 times, and the distribution of the proportion of neurons with event-related activity at the stage was obtained. The cumulative probability against the proportion was calculated using the cumulative distribution function (cdfplot; MATLAB, MathWorks, Natick, MA).

Licking responses were detected with an infrared photobeam sensor placed in the 3D printed reward port and generated timestamps for the onset of each spout contact according to the analog voltage signal. The licking responses corresponding to four kinds of trial in the first licking-related neurons were shown as averaged rates (50-ms bins) aligned based on the timing of the choice response. To calculate the correlation coefficient between neuronal activity and licking responses, the spikes and licking for 5 s after the first licking were counted every trial for each first licking-related neuron. Trials in which neurons never responded were removed from both the spike and licking data. The correlation coefficients between the two kinds of data were calculated for each neuron (corr; MATLAB, MathWorks), and neurons showing significant differences were determined by applying the Pearson correlation coefficient (p < 0.01).

## Histology

After the behavioral experiments, rats were anesthetized with pentobarbital (50 mg/kg body weight) and isoflurane (3%), then transcardially perfused with 4% paraformaldehyde in phosphate-buffer (PB). The brains were post-fixed overnight at 4 °C and cryoprotected with 10%, 20%, and 30% sucrose in PB.

Brain tissues were cut into 30-µm sections by using a cryostat (CM3050 S, Leica Biosystems, Wetzlar, Germany). To define the lesioned areas in the striatum, sections were immunostained with a mouse anti-NeuN antibody (1:1000 dilution; MAB377, Millipore, Burlington, MA), and then with a goat anti-mouse IgG antibody conjugated with Alexa Fluor 488 (1:1000 dilution; A-11029, Thermo Fisher Scientific, Waltham, MA). The sections were counterstained with 0.3 µM 4',6-diamidino-2-phenylindole (DAPI; 0.1 µg/mL, D1306, Thermo Fisher Scientific). For the transient inactivation experiment, the positions of guide cannulae were confirmed by staining with 0.1% cresyl violet solution (41021, Muto Pure Chemicals, Tokyo, Japan). For the electrophysiological experiments, small lesions were made by applying an electrolytic current (3 µA, 10 s) through top and bottom channels of each shank by using the stimulus isolator (A365, World Precision Instruments, Sarasota, FL). Sections through the striatum were stained with a mouse anti-TH antibody (1:1000 dilution; MAB318-AF488, Millipore), and then with a goat anti-mouse IgG antibody conjugated with Alexa Fluor 488 (1:1000 dilution; Jackson Immunoresearch Laboratories, West Grove, PA). The recording sites were verified by staining with DAPI and red fluorescent Nissl stain solution (1:10000/1:200 dilution; N21482, Thermo Fisher Scientific). For the tracing, sections through the ventral midbrain were stained with a mouse anti-TH antibody (1:1000 dilution), and then with a donkey anti-mouse IgG antibody conjugated with Alexa Fluor 647 (1:1000 dilution; A-31571, Thermo Fisher Scientific). Fluorescence and immunostained signals were visualized under fluorescence microscopes (BZ-X800, Keyence, Osaka, Japan) and confocal microscope (A1R, Nikon, Osaka, Japan).

## Statistical analysis

Statistical analysis was performed using MATLAB (MathWorks) and SPSS Statistics (version 27; IBM, Armonk, NY). Comparisons of data between two groups were performed using the Student's $t$-test (paired), the Wilcoxon signed rank test (paired), and the Wilcoxon rank-sum test (unpaired). Comparisons of data among three groups were constructed using a one-way repeated measures ANOVA, two-way repeated measures ANOVA, or the Kruskal-Wallis test (without assuming normal distribution and homogeneity of variances). All statistical tests were two-sided. The Bonferroni post-hoc test and Mann–Whitney U-test with Bonferroni adjustment were carried out as needed to compare means. For boxplots, the data points below the lower (25th percentile) −1.5 interquartile range and above the upper (75th percentile) quartiles +1.5 interquartile range were regarded as outliers, and they are not included in the boxplots.

## Acknowledgements

We thank Drs. Keiji Ota, Hikaru Yokoyama, and Sho Aoki for instructing the behavioral analysis and Dr. Hideyuki Matsumoto for technical advice and discussion. We thank the Research Support Platform, Osaka Metropolitan University Graduate School of Medicine for microscopic imaging. This work was supported by grants-in-aid for JSPS Fellows (19J01997, SS), Scientific Research (C) (21K11556, SS), and Scientific Research (B) (25K02365, KK) from the Japan Society for the Promotion of Science; a grant-in-aid for Scientific Research on Transformative Research Areas (A) Adaptive Circuit Census (21H05244, KK) from the Ministry of Education, Science, Sports, and Culture of Japan; the Grant for Casio Science Promotion Foundation, Takeda Science Foundation; and the Nakatomi Foundation (SS).

## Additional information

### Funding

| Funder | Grant reference number | Author |
| --- | --- | --- |
| Japan Society for the Promotion of Science | 19J01997 | Susumu Setogawa |
| Japan Society for the Promotion of Science | 21K11556 | Susumu Setogawa |
| Japan Society for the Promotion of Science | 25K02365 | Kazuto Kobayashi |

| Funder | Grant reference number | Author |
|---|---|---|
| Ministry of Education, Science, Sports, and Culture of Japan | 21H05244 | Kazuto Kobayashi |
| Casio Science Promotion Foundation | | Susumu Setogawa |
| Takeda Science Foundation | | Susumu Setogawa |
| Nakatomi Foundation | | Susumu Setogawa |

The funders had no role in study design, data collection and interpretation, or the decision to submit the work for publication.

## Author contributions

Susumu Setogawa, Conceptualization, Formal analysis, Funding acquisition, Investigation, Methodology, Writing – original draft, Writing – review and editing; Takashi Okauchi, Di Hu, Yasuhiro Wada, Keigo Hikishima, Hirotaka Onoe, Kayo Nishizawa, Nobuyuki Sakayori, Hiroyuki Miyawaki, Takuma Kitanishi, Kenji Mizuseki, Yilong Cui, Investigation; Kazuto Kobayashi, Conceptualization, Supervision, Funding acquisition, Validation, Investigation, Visualization, Writing – original draft, Project administration, Writing – review and editing

## Author ORCIDs

Susumu Setogawa (ID) https://orcid.org/0000-0002-9161-2656
Kenji Mizuseki (ID) https://orcid.org/0000-0002-1456-2149
Kazuto Kobayashi (ID) https://orcid.org/0000-0002-7617-2939

## Ethics

All procedures for animal care and handling were approved by the Institutional Animal Care and Use Committees of RIKEN Center for Biosystems Dynamics Research, Fukushima Medical University, and Osaka City University. Animal procedures were carried out in accordance with the guidelines established by their Institutional Committees. All efforts were made to minimize the number of animals used and their suffering throughout the course of the experiments.

Reviewer #1 (Public review): https://doi.org/10.7554/eLife.97326.4.sa1
Reviewer #2 (Public review): https://doi.org/10.7554/eLife.97326.4.sa2
Author response https://doi.org/10.7554/eLife.97326.4.sa3

---

# Additional files

## Supplementary files

MDAR checklist

Source data 1. Behavioral strategies on Day N-1 of rats used for the injection into striatal subregions. xlsx.

## Data availability

All data generated or analysed during this study are included in the manuscript and supporting files; source data files have been provided for Figures 1-7 and related figure supplements. The dataset is also deposited on Mendeley data.

The following dataset was generated:

| Author(s) | Year | Dataset title | Dataset URL | Database and Identifier |
|---|---|---|---|---|
| Kobayashi K, Setogawa S | 2025 | Acquisition of Auditory Discrimination Mediated by Different Processes through Two Distinct Circuits Linked to the Lateral Striatum | https://doi.org/10.17632/ghzr838bdk.3 | Mendeley Data, 10.17632/ghzr838bdk.3 |

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
