## [Editor Report · eLife Assessment]

This study provides an **important** understanding of the contribution of different striatal subregions, the anterior Dorsal Lateral Striatum (aDLS) and the posterior Ventrolateral Striatum (pVLS), to auditory discrimination learning. The authors have included robust behavior combined with multiple observational and perturbation techniques. The data provided are **convincing** of the relevance of task-related activity in these two subregions during learning.

---

## [Referee Report · Reviewer #1 (Public review)]

In this study, Setogawa et al. employ an auditory discrimination task in freely moving rats, coupled with small animal imaging, electrophysiological recordings, and pharmacological inhibition/lesioning experiments to better understand the role of two striatal subregions: the anterior Dorsal Lateral Striatum (aDLS) and the posterior Ventrolateral Striatum (pVLS), during auditory discrimination learning. Attempting to better understand the contribution of different striatal subregions to sensory discrimination learning strikes me as a highly relevant and timely question, and the data presented in this study are certainly of major interest to the field. The authors have set up a robust behavioral task, systematically tackled the question about a striatal role in learning with multiple observational and manipulative techniques. Additionally, the structured approach the authors take by using neuroimaging to inform their pharmacological manipulation experiments and electrophysiological recordings is a strength.

---

## [Referee Report · Reviewer #2 (Public review)]

The study by Setogawa et al. aims to understand the role that different striatal subregions belonging to parallel brain circuits have in associative learning and discrimination learning (S-O-R and S-R tasks). Strengths of the study are the use of multiple methodologies to measure and manipulate brain activity in rats, from microPET imaging to excitotoxic lesions and multielectrode recordings across anterior dorsolateral (aDLS), posterior ventral lateral (pVLS), and dorsomedial (DMS) striatum.

---

## [Author Response]

The following is the authors’ response to the previous reviews.

Although we have no further revisions on the manuscript, we would like to respond to the remaining comments from the reviewers as follows.

**Reviewer 1:**
The authors have addressed some concerns raised in the initial review but some remain. In particular it is still unclear what conclusions can be drawn about taskrelated activity from scans that are performed 30 minutes after the behavioral task. I continue to think that a reorganization/analysis data according to event type would be useful and easier to interpret across the two brain areas, but the authors did not choose to do this. Finally, switching the cue-response association, I am convinced, would help to strengthen this study.

As for the task-related activity, the strategy for PET scan was explained in our response to the comment 2 from Reviewer 2. Briefly, rats receive intravenous administration of 18F-FDG solution before the start of the behavioral session. The 18FFDG uptake into the cells starts immediately and reaches the maximum level until 30 min, being kept at least for 1 h. A 30-min PET scan is executed 25 min after the session. Therefore, the brain activity reflects the metabolic state during task performance in rats.

Regarding data presentation of the electrophysiological experiments, we described the subpopulations of event-related neurons showing notable neuronal activity patterns in the order of aDLS and pVLS, according to the procedure of explanations for the behavioral study

For switching the cue-response association, we mentioned the difference in firing activity between HR and LL trials, suggesting that different combinations between the stimulus and response may affect the level of firing activity. As suggested by the reviewer, an examination of switching the cue-response association is useful to confirm our interpretation. We will address this issue in our future studies.

**Reviewer 2:**
The authors have made important revisions to the manuscript and it has improved in clarity. They also added several figures in the rebuttal letter to answer questions by the reviewers. I would ask that these figures are also made public as part of the authors' response or if not, included in the manuscript.

We will present the figures publicly available as part of our response.